# Large-scale filament formation inhibits the activity of CTP synthetase

Rachael M Barry[1], Anne-Florence Bitbol[2], Alexander Lorestani[1], Emeric J Charles[3], Chris H Habrian[4], Jesse M Hansen[3], Hsin-Jung Li[1], Enoch P Baldwin[4], Ned S Wingreen[1,2], Justin M Kollman[3,5], Zemer Gitai[1]*

[1]Department of Molecular Biology, Princeton University, Princeton, United States; [2]Lewis-Sigler Institute for Integrative Genomics, Princeton University, Princeton, United States; [3]Department of Anatomy and Cell Biology, McGill University, Montreal, Canada; [4]Department of Molecular and Cellular Biology, University of California, Davis, Davis, United States; [5]Groupe de Recherche Axé sur la Structure des Protéines, McGill University, Montreal, Canada

**Abstract** CTP Synthetase (CtpS) is a universally conserved and essential metabolic enzyme. While many enzymes form small oligomers, CtpS forms large-scale filamentous structures of unknown function in prokaryotes and eukaryotes. By simultaneously monitoring CtpS polymerization and enzymatic activity, we show that polymerization inhibits activity, and CtpS's product, CTP, induces assembly. To understand how assembly inhibits activity, we used electron microscopy to define the structure of CtpS polymers. This structure suggests that polymerization sterically hinders a conformational change necessary for CtpS activity. Structure-guided mutagenesis and mathematical modeling further indicate that coupling activity to polymerization promotes cooperative catalytic regulation. This previously uncharacterized regulatory mechanism is important for cellular function since a mutant that disrupts CtpS polymerization disrupts *E. coli* growth and metabolic regulation without reducing CTP levels. We propose that regulation by large-scale polymerization enables ultrasensitive control of enzymatic activity while storing an enzyme subpopulation in a conformationally restricted form that is readily activatable.

**\*For correspondence:** zgitai@princeton.edu

**Reviewing editor**: Mohan Balasubramanian, University of Warwick, United Kingdom

## Introduction

Many enzymes form small-scale oligomers with well-defined subunit numbers, typically ranging from 2 to 12 subunits per oligomer. Recent studies suggest that some enzymes can also form large, higher-order polymers in which dozens to hundreds of subunits assemble into filaments (*Barry and Gitai, 2011*). For most of these structures, we lack an understanding of both the regulation and functional significance of their polymerization. To address these questions, we focused on the assembly of CTP synthetase (CtpS), an essential and universally conserved metabolic enzyme. CtpS forms large, micron-scale filaments in a wide variety of bacterial and eukaryotic species (*Ingerson-Mahar et al., 2010*; *Liu, 2010*; *Noree et al., 2010*), but the structure of these polymers, what triggers their formation, and the relationship between CtpS polymerization and enzymatic activity were unknown until now.

Cellular CTP levels are subject to exquisitely tight homeostatic control, and CtpS is one of the most regulated enzymes in the cell. In both prokaryotes and eukaryotes, CtpS activity is regulated by allosteric control and feedback-inhibition of enzymatic activity, and CtpS levels are regulated by transcriptional and post-translational control (*Long and Pardee, 1967*; *Levitzki and Koshland, 1972b*; *Yang et al., 1996*; *Meng et al., 2004*). Cells in all kingdoms of life synthesize CTP using CtpS (*Long and Pardee, 1967*), and its essentiality makes CtpS an attractive chemotherapeutic and antiparasitic target (*Williams et al., 1978*; *Hofer et al., 2001*).

**eLife digest** Enzymes are proteins that perform reactions that can convert one or more chemicals (the substrates) into others (the products). The rate at which an enzyme produces its product is often carefully regulated. Some molecules slow or stop an enzyme by binding to and blocking the site where its substrates normally bind: its 'active site'. Other molecules can also bind to sites other than the active site, which can cause the enzyme to become either more or less active.

Almost all living things have an enzyme called CTP synthetase that makes one of the building blocks that is used to build DNA and a similar molecule called RNA. This enzyme converts a molecule called uridine triphosphate (or UTP) into another called cytidine triphosphate (CTP): a reaction that is powered by breaking down molecules of adenosine triphosphate (ATP).

The amount of CTP synthetase made by a cell is carefully controlled. The enzyme's activity is also regulated by the levels of UTP and CTP, and by another molecule (called GTP) that binds to a site outside of its active site. Four copies of the CTP synthetase protein must work together before this enzyme can turn UTP into CTP. The enzyme also forms much larger aggregates, or polymers; however, it is not clear what causes these polymers to form, or what they do in a cell.

Barry et al. have now discovered that CTP synthetase is almost completely inactivated when these polymers are formed. Furthermore, CTP encourages the polymers to form, whilst UTP and ATP cause them to disassemble. Therefore, this enzyme is least active when there is excess product in the cell, and most active when its substrates are plentiful.

By determining the three-dimensional structure of a CTP synthetase polymer, Barry et al. reveal that although CTP is bound to the enzymes, their active sites are still freely accessible. However, the enzymes in the polymer appear to be locked into a shape that makes them unable to carry out their function. When Barry et al. then mutated the enzyme so that it was unable to form polymers it was also no longer inactivated in the same way by CTP. Bacterial cells with only these mutant versions of CTP synthetase are unable to properly control their levels of CTP. This suggests that polymer formation is important for regulating this enzyme in response to a build up of its product. Further work is needed to see whether the regulation of CTP synthetase activity by forming polymers is specific to this enzyme or a widespread mechanism that is used to control other enzymes too.

The CtpS enzyme has two domains connected by an elongated linker: a glutaminase (GATase) domain that deaminates glutamine and a synthetase (ALase) domain that aminates UTP in an ATP-dependent manner to form CTP. CtpS has binding sites for substrates (glutamine, ATP, and UTP), product (CTP), and a proposed binding site for an allosteric modulator (GTP) (**Levitzki and Koshland, 1972b**). CtpS tetramerization is necessary for its catalytic activity and is controlled by nucleotide availability; ATP, UTP, or CTP can favor tetramer formation (**Figure 1A**; **Levitzki and Koshland, 1972a**; **Anderson, 1983**; **Pappas et al., 1998**; **Endrizzi et al., 2004**). Of critical regulatory importance, CtpS activity is also inhibited by CTP (**Long and Pardee, 1967**).

Here, we determine the function and mechanism of CtpS polymerization. We demonstrate that CtpS polymerization negatively regulates CtpS activity when its CTP product accumulates. We also present the structure of the CtpS polymers and the resulting implications for CtpS inhibition. We confirm the physiological significance of CtpS assembly by demonstrating that polymerization-mediated regulation is essential for the proper growth and metabolism of *Escherichia coli*. Together, these findings establish CtpS as a model for understanding enzymatic regulation by large-scale polymerization. Finally, we model how coupling CtpS activity to its large-scale assembly can enable cooperative regulation and discuss the implications of polymerization-based regulation for ultrasensitive metabolic control and cytoskeletal evolution.

## Results

### CtpS polymerization inhibits enzymatic activity

Because CtpS filament formation is conserved between divergent organisms, we hypothesized that CtpS polymerization may regulate its conserved enzymatic function. We therefore designed a system to simultaneously monitor the assembly and activity of purified *E. coli* CtpS. We used a fluorometer to assay CtpS assembly by right-angle light scattering and CtpS activity by the specific absorbance of its

CTP product. CtpS assembly and activity were assayed across a range of enzyme concentrations in activity buffer containing saturating amounts of substrates (UTP, ATP, and glutamine) as well as GTP and $Mg^{2+}$ (referred to as 'activity buffer' throughout the text) (**Figure 1B**). CtpS protein was first pre-incubated in an incomplete activity buffer without glutamine to favor active tetramer formation. CTP production was then initiated by the addition of glutamine to form a complete activity buffer. The formation of well-ordered filaments was confirmed by negative stain electron microscopy (EM) (**Figure 1C**). Interestingly, at CtpS levels where robust changes in light scattering are observed (above approximately 1–2 µM), CtpS activity (determined by the rate of CTP production per enzyme) sharply decreases (**Figure 1B**, **Figure 1—figure supplements 1 and 2**). This abrupt transition in activity state supports the hypothesis that there is a threshold for polymerization and that polymerization is inhibitory. Noise and nonlinearity in the light scattering data make it difficult to determine an exact critical concentration value. However, based on correlation between light scattering and CTP production changes, we predict the

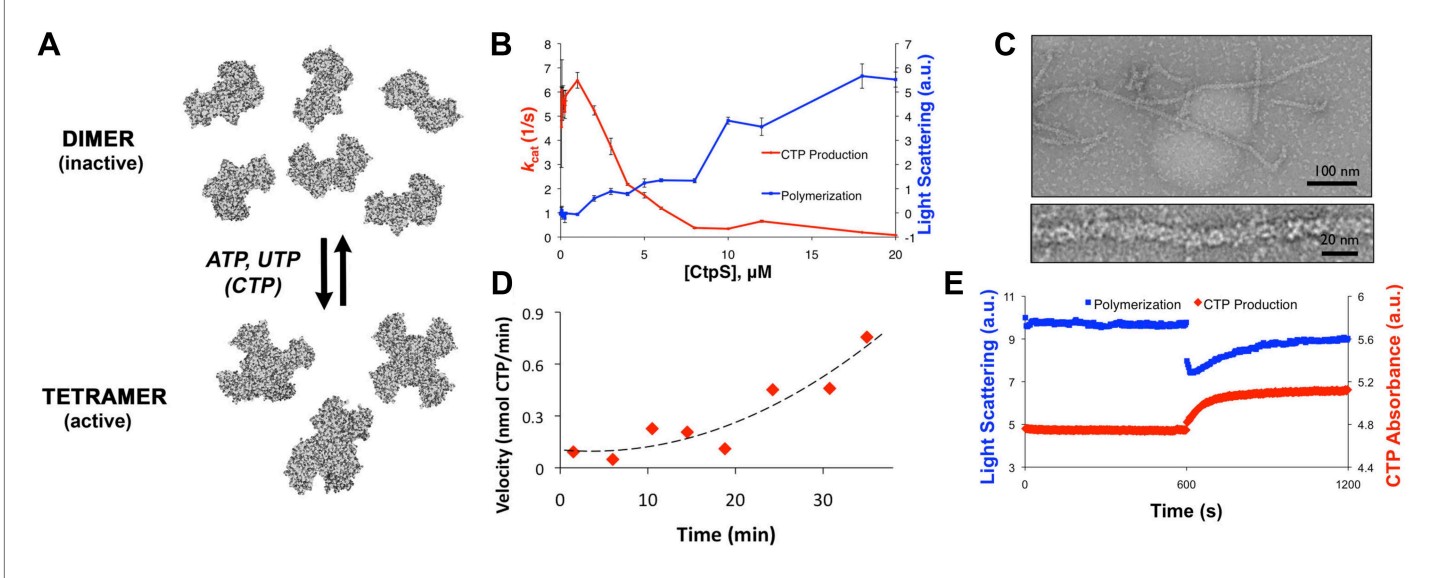

**Figure 1.** CtpS polymerization and enzymatic activity are inversely related. (**A**) A model of oligomeric regulation of CtpS. Tetramer formation from CtpS dimers is favored by a combination of enzyme concentration as well as nucleotide (substrates ATP and UTP or product CTP) and $Mg^{2+}$ binding. (**B**) CtpS was incubated in activity buffer containing all substrates for CTP production. As the enzyme concentration increases, CtpS shows assembly by light scattering and the $k_{cat}$ value ($V_{obs}$/[CtpS]) decreases. Error bars = standard error (SE), n = 3–5. (**C**) Negative stain image of CtpS filaments assembled after CTP synthesis reaction. Smaller particles in the background resemble the X-shaped CtpS tetramer. A single filament is shown at bottom. (**D**) CtpS polymers formed in activity buffer were ultracentrifuged to pellet polymers. The pellet fraction was resuspended and CTP production was recorded. (**E**) CtpS assembly and activity were assayed after CtpS was first polymerized, followed by addition of saturating amounts of substrate after 600 s.

The following figure supplements are available for figure 1:

**Figure supplement 1**. Determination of threshold concentration for CtpS polymerization in activity buffer.

**Figure supplement 2**. Representative examples of raw data from three different concentrations of CtpS incubated in activity buffer included in **Figure 1B**.

**Figure supplement 3**. Calculation of intracellular CtpS in minimal media.

**Figure supplement 4**. CtpS activity is not sensitive to incubation on ice.

**Figure supplement 5**. CtpS higher order structures disassemble over time after centrifugation.

**Figure supplement 6**. CtpS polymer disassembly is not caused by mechanical disruption of polymers.

**Figure supplement 7**. Correction of kcat values between initial Princeton and UC Davis data sets.

assembly threshold of CtpS to be approximately 1–2 µM. The cellular level of CtpS protein in *E. coli* grown in minimal media was measured at 2.3 µM (*Figure 1—figure supplement 3*), indicating that the CtpS polymerization observed in vitro may be physiologically favorable.

To determine if polymerization indeed inhibits CtpS activity, we assayed the activity of polymers purified by ultracentrifugation. The polymer-containing pellet was least enzymatically active immediately after centrifugation and CtpS activity increased as the polymers in the pellet disassembled (*Figure 1D*, *Figure 1—figure supplements 4 and 5*). CtpS polymers are thus inactive or much less than maximally active and polymerization is readily reversible. We directly demonstrated the reversibility of CtpS assembly and inactivation by first allowing CtpS to polymerize in activity buffer (with all substrates present) and then adding 1 mM UTP and ATP. Upon addition of these substrate nucleotides, we observed a sharp decrease in light scattering that corresponded to a sharp increase in CtpS activity. This transition was followed by a gradual increase in light scattering and corresponding decrease in activity back to the initial residual level (*Figure 1E*). Control experiments confirmed that the decrease in CtpS polymerization was not due to mechanical disruption by substrate addition (*Figure 1—figure supplement 6*). The correlation between the decrease in light scattering and the initiation of CTP production at the time of substrate addition indicates that substrate addition leads to rapid depolymerization and subsequent enzyme reactivation. Immediately after this point, we observed an increase in both CTP levels and polymerization. We therefore conclude that polymerized CtpS enzymes are inactive and must disassociate from the polymer to resume normal enzymatic activity. Despite the fact that polymerization occurs in a buffer containing substrates, polymerization only occurs with CTP production, suggesting that polymerization is triggered not by the initial substrates, but rather by the accumulation of CTP product.

## CtpS polymerization is induced by its product and repressed by its substrate

In order to identify the factors that control CtpS inhibition by assembly, we first confirmed that none of the substrates alone induced polymerization (*Figure 2—figure supplement 1*). We then directly tested our hypothesis that CtpS's product, CTP, a known inhibitor of CtpS activity, stimulates CtpS polymerization. In the absence of substrates (UTP, ATP, and glutamine), incubation with CTP caused CtpS to polymerize (*Figure 2A*). The threshold concentration for robust changes in light scattering by CtpS with saturating CTP (1–2 µM CtpS; *Figure 2—figure supplement 2*) agrees with the threshold concentration in the presence of substrates (1–2 µM CtpS; *Figure 1—figure supplement 1*). This result suggests that CTP alone is sufficient to influence polymerization and that the substrates and any other products of the enzymatic reaction are not necessary. To confirm that CTP stimulates CtpS assembly, we used ultracentrifugation as an independent assembly assay. Titrating with increasing amounts of CTP caused an increase in the amount of CtpS found in the pellet with respect to the 0 mM CTP condition (*Figure 2B*, *Figure 2—figure supplement 3*).

We further demonstrated that CTP binding is necessary for polymerization by showing that a CtpS[E155K] mutant defective for CTP-binding feedback inhibition (reviewed in *Endrizzi et al., 2005*) (*Trudel et al., 1984*; *Ostrander et al., 1998*) fails to polymerize under the same CTP-producing conditions in which wild-type enzyme polymerizes (*Figure 2C*). Furthermore, electron microscopy confirmed that, unlike wild-type CtpS, CtpS[E155K] does not polymerize in the presence of CTP (*Figure 2D*). Together, our data indicate that within our studied range of enzyme concentrations, CtpS's product, CTP, is both necessary and sufficient to induce CtpS polymerization.

The CtpS crystal structure suggests that the enzyme's UTP and CTP binding sites partially overlap (*Endrizzi et al., 2005*), raising the question of whether CtpS assembly is controlled by the absolute level of CTP or the relative product/substrate levels. 6-Diazo-5-oxo-L-norleucine (DON) is a glutamine analog that covalently binds glutaminase active sites and irreversibly inactivates enzymatic activity (*Chakraborty and Hurlbert, 1961*). When added to activity buffer, DON abolishes both CTP production and CtpS polymerization (*Figure 2—figure supplement 4*). However, DON-treated CtpS can still polymerize when CTP is added to the solution (*Figure 2E*). Polymers formed in the presence of CTP and DON disassemble upon the addition of substrates but do not reform after substrate addition (*Figure 2E*), presumably because the DON-inhibited CtpS cannot produce additional CTP. DON treatment has no effect on CtpS polymerization when the enzyme is incubated with saturating CTP (*Figure 2—figure supplements 1 and 5*). These results suggest that competition between substrate (UTP) and product (CTP) binding controls the polymerization equilibrium of CtpS. The dependence of

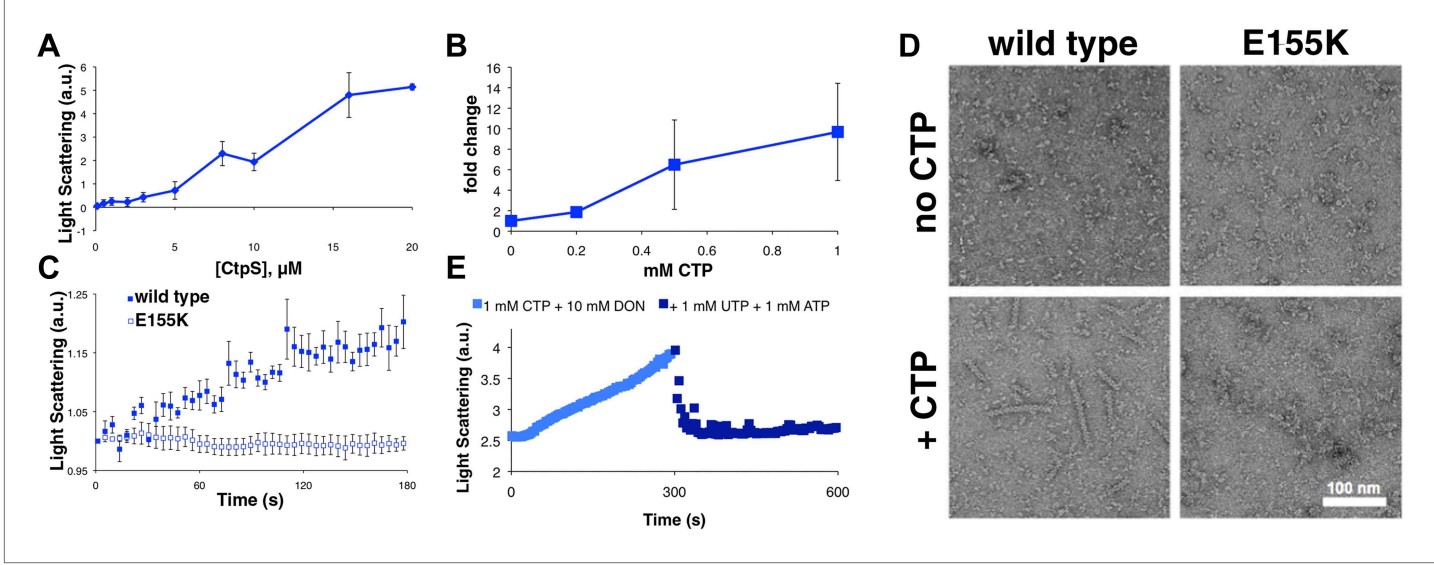

**Figure 2**. CTP is sufficient and necessary to stimulate CtpS polymerization. (**A**) CtpS levels were titrated in buffer containing 1 mM CTP (with no substrates present). Polymerization was observed in the same range of protein concentrations as in activity buffer. Error bars = SE, n = 3. (**B**) CtpS was allowed to polymerize at different CTP concentrations (with no substrates present). The polymers were collected by ultracentrifugation and changes in CtpS pellet abundance were quantified by immunoblot. Error bars = SE, n = 2. (**C**) Purified CtpS[E155K], which is defective in CTP binding, showed no obvious changes in light scattering during the normal conditions of wild-type polymer assembly in activity buffer. Initial light scattering values were normalized to 1 to place wild-type CtpS and CtpS[E155K] on the same scale. Error bars = SE, n = 3. (**D**) CtpS Filaments of wild-type and mutants by negative stain electron microscopy. There were very few filaments observed in the absence of CTP (top row). Upon the addition of nucleotide and MgCl$_2$, filaments were only observed in the wild-type sample (first column). Micrographs were all taken at 55,000X magnification. (**E**) CtpS was incubated in the inhibitor DON and 1 mM CTP and allowed to polymerize. Addition of ATP and UTP depolymerized the sample. Polymers did not reform.

The following figure supplements are available for figure 2:

**Figure supplement 1**. CtpS enzymatic activity or CTP addition is required for CtpS polymerization.

**Figure supplement 2**. Determination of threshold concentration for CtpS polymerization in 1 mM CTP.

**Figure supplement 3**. Immunoblot of CtpS pelleted by ultracentrifugation.

**Figure supplement 4**. DON-treated CtpS is enzymatically inactive.

**Figure supplement 5**. DON inhibition of activity does not inhibit polymerization upon CTP addition.

polymerization on CTP levels may explain why DON treatment abolishes in vivo CtpS assembly in some cellular contexts (*Ingerson-Mahar et al., 2010*) but not others (*Chen et al., 2011*).

## The structure of the CtpS polymer suggests a mechanism for enzymatic inhibition

To better understand the mechanism of enzymatic inhibition by polymerization, we determined the structure of the CtpS filament by cryo-electron microscopy at 8.4 Å resolution (*Figure 3—figure supplement 1*). The repeating subunits of the filament are X-shaped CtpS tetramers (*Figure 3A*). The helical symmetry of the filament results in CtpS tetramers stacked atop one another with the arms of the adjacent Xs interdigitated. The 222 point group symmetry of the tetramer is maintained within the filament, resulting in overall twofold symmetry both along and perpendicular to the helical axis. A significant effect of this unusual symmetry is that, unlike many biological polymers, CtpS filaments are apolar.

To create an atomic model of the CtpS filament, we fit a monomer of the *E. coli* CtpS crystal structure into the cryo-EM structure as three rigid bodies (ALase domain, GATase domain, and the linker region) (*Figure 3B*). There is a slight rotation between the GATase and ALase domains, similar to the

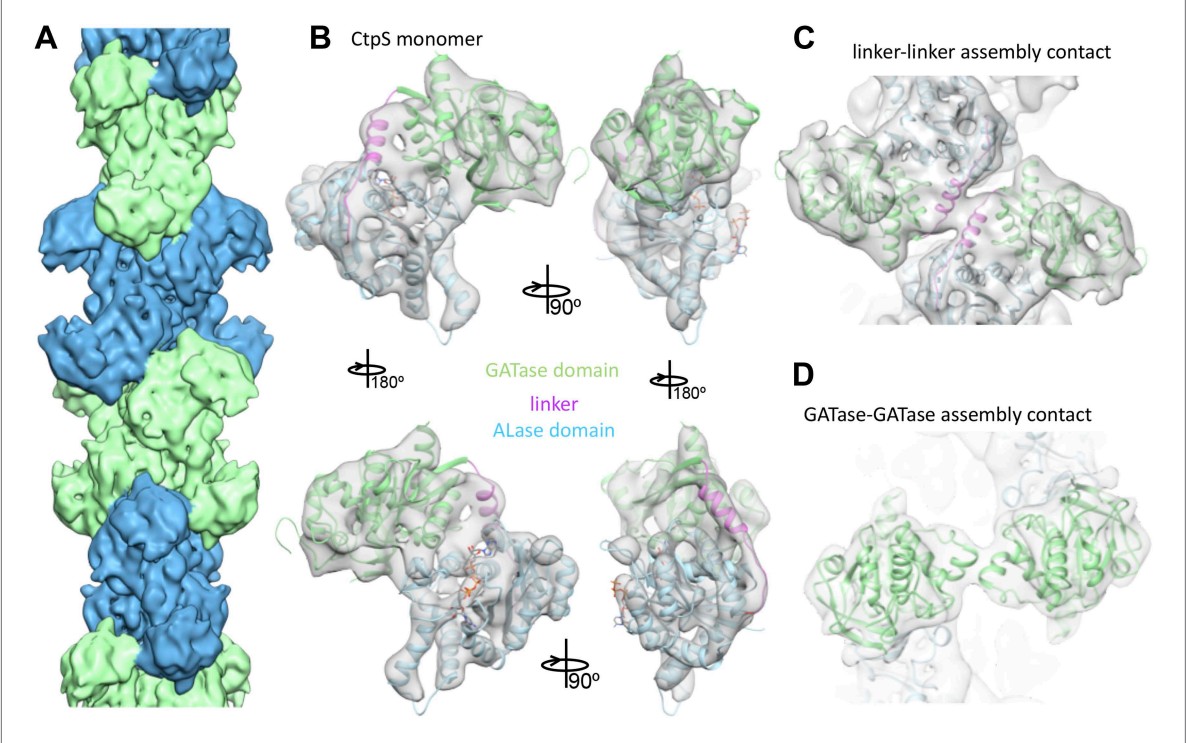

**Figure 3**. Cryo-EM structure of CtpS filaments at 8.4 Å resolution. (**A**) A segment of the reconstructed filament, colored by helical subunit. (**B**) The *E. coli* CtpS crystal structure monomer fit into the cryo-EM density. Each domain was fit as a separate rigid body. (**C**) Novel filament assembly contacts between the linker domains. (**D**) Novel assembly contacts between the GATase domains.

The following figure supplements are available for figure 3:

**Figure supplement 1**. Cryo-EM reconstruction of CtpS filaments.

**Figure supplement 2**. The CtpS monomer in the filament is in a similar conformation to crystallographic structures, and ADP and CTP are present.

variation seen across crystal structures of full length CtpS (*Figure 3—figure supplement 2A*). There is a strong density for CTP bound at the inhibitory site, and no density in the predicted UTP active site (*Figure 3—figure supplement 2B*), confirming the biochemical data that CTP binding favors assembly. Weaker density is also observed for ADP, but there is no density in the predicted GTP allosteric regulatory site (*Figure 3—figure supplement 2C,D*). There is a minor rearrangement of the tetramerization interface in the filament relative to the crystal structure that results in a compression of the tetramer by about 3 Å along the length of the filament axis (*Figure 4*).

The cryo-EM structure of the CtpS filament offers insight into the mechanism of enzymatic regulation. All of the enzyme active sites are solvent accessible, suggesting that UTP, ATP, and glutamine can freely diffuse into the filament (*Figure 5A*). This observation rules out occlusion of active sites as a regulatory mechanism. An alternative mechanism of CtpS inhibition is blocking the transfer of ammonia between the GATase and ALase active sites, which are separated by ~25 Å. The detailed mechanism of ammonia transfer is unknown, but likely involves a conformational rearrangement in the vicinity of a putative channel that connects the two domains (*Endrizzi et al., 2004*; *Goto et al., 2004*). One prediction is that a conformational change, induced by UTP and ATP binding, rotates the GATase domain toward the ALase domain to create a shorter channel between the active sites (*Goto et al., 2004*). Such a large-scale rotation would be unattainable in the steric environment of the filament, as it would lead to clashing of the moving GATase domain with an adjacent CtpS tetramer (*Figure 5B,C*). Regardless of the specific changes involved, quaternary constraints imposed by the filament structure likely provide the mechanism for inhibition of the synthesis reaction.

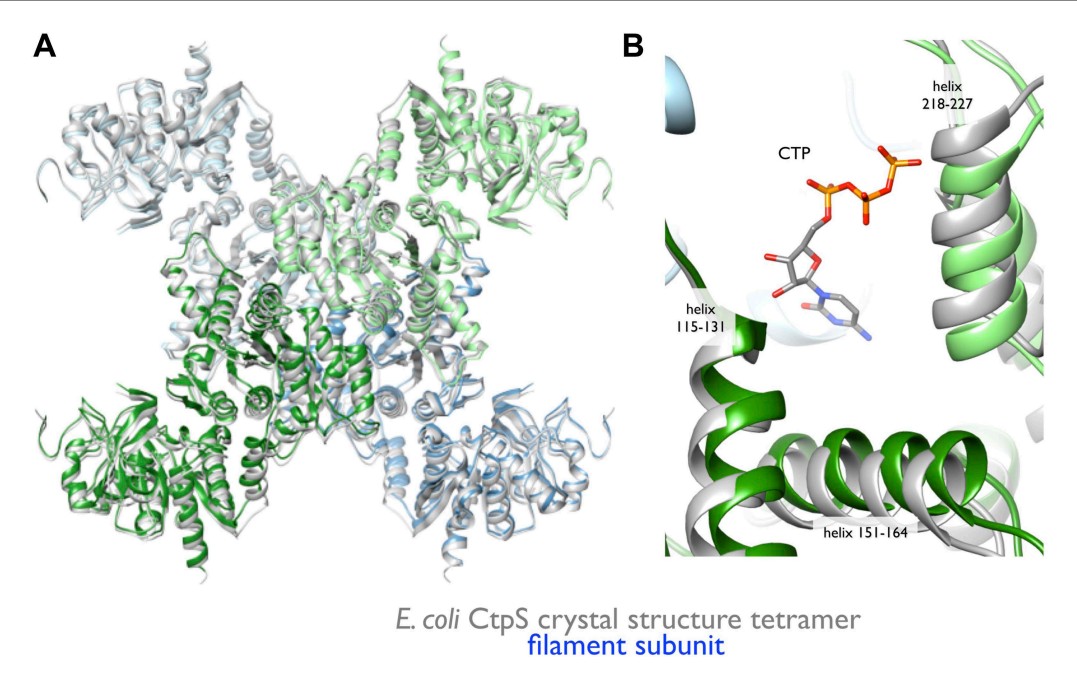

**Figure 4**. Rearrangement of the CtpS tetramerization interface within the filament. (**A**) Superposition of the *E. coli* crystallographic tetramer (gray) with the atomic model from the cryo-EM structure (color), shows a rearrangement of the tetramerization contacts, primarily a compression of the tetramer along the filament axis. (**B**) Rearrangements of the tetramerization contacts shift the relative positions of helices near bound CTP (gray: crystal structure; color cryo-EM structure).

## A CtpS polymerization interface mutant disrupts feedback regulation

To validate the filament structure and its mechanistic implications, we generated structure-guided mutants in the CtpS polymerization interface. Two discrete segments constitute the novel filament assembly contacts: the linker region α-helix 274–284, and the short α-helix 330–336 of the GATase domain (*Figure 3D,E*). Though the exact amino acid sequences at the inter-tetramer assembly interfaces are not well conserved, relative to the rest of CtpS, both sites feature many charged or hydrophobic residues available for potential polymerization stabilization across species (*Figure 6—figure supplement 1*). We previously demonstrated that in *E. coli*, an mCherry-CtpS fusion faithfully reproduces the filamentous localization of native CtpS (as assayed by immunofluorescence) (*Ingerson-Mahar et al., 2010*). As an initial screen for CtpS assembly, we therefore introduced four mutations in the linker region α-helix and surrounding residues (E277R, F281R, N285D, and E289R) into mCherry-CtpS (*Figure 6A*). All four polymerization interface mutants disrupted mCherry-CtpS localization, exhibiting a diffuse localization pattern rather than linear filaments (*Figure 6B*).

The loss of filamentous mCherry-CtpS localization does not exclude the possibility that the polymerization interface mutants form small filaments that cannot be resolved by light microscopy. Consequently, to determine if the diffuse localization in vivo reflected a polymerization defect, we purified one of the linker region helix mutants, CtpS$^{E277R}$, and examined its polymerization by light scattering and EM. CtpS$^{E277R}$ did not significantly polymerize in activity buffer, and no filaments could be detected by EM (*Figure 7B*, *Figure 7—figure supplement 1*), confirming that CtpS$^{E277R}$ cannot properly polymerize. We attribute the slight linear increase in light scattering with increasing concentration of CtpS$^{E277R}$ to the increase in protein abundance.

We next determined the impact of the E277R polymerization interface mutation on CtpS activity. At the lowest protein concentration tested, CtpS$^{E277R}$ exhibited slightly reduced CTP production (71% of wild type maximal activity) compared to the wild type protein (*Figure 7A*). To determine if the polymerization defect of CtpS$^{E277R}$ was due to impaired large-scale assembly or reduced CTP production, we used EM to examine its polymerization in the presence of saturating CTP levels. CtpS$^{E277R}$ did not

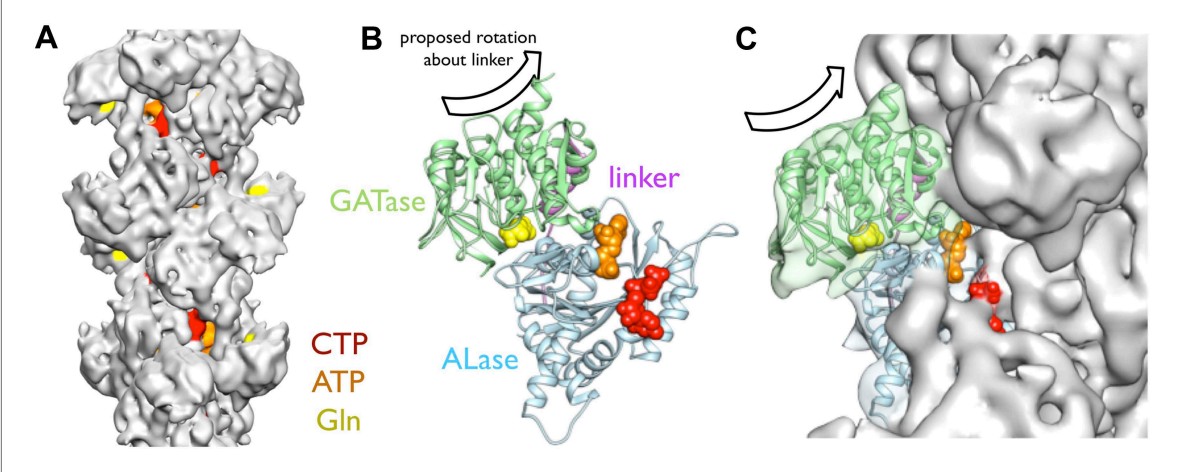

**Figure 5**. Implications of the CtpS filament structure for the mechanism of enzyme inhibition. (**A**) The binding sites for ATP, CTP, and glutamine are all solvent accessible in the filament, suggesting that they are freely exchangeable in the filament form. (**B**) The approximate direction of the putative rotation of the glutaminase domain toward the amidoligase domain (arrow), which is predicted to create a shorter channel for ammonia diffusion. (**C**) In the filament structure, such a conformational change would be sterically hindered by contacts with adjacent filament subunits.

polymerize in the presence of high levels of CTP (*Figure 7B*). We thus conclude that CtpS$^{E277R}$ impairs polymerization independently of its effect on activity.

Whereas CtpS$^{E277R}$ was slightly impaired in its activity at low enzyme concentrations, CtpS$^{E277R}$ exhibited a much higher concentration at which $k_{cat}$ is one half of its maximum due to polymerization (the $[CtpS]_{0.5}$ value) compared to wild-type CtpS ($[CtpS^{E277R}]_{0.5}$ = 7.1 µM vs $[CtpS]_{0.5}$ = 3.3 µM). Furthermore, the concentration dependence of CtpS$^{E277R}$ $k_{cat}$ was less steep than wild type, with CtpS$^{E277R}$ retaining 48% of its maximal activity at the highest enzyme concentration tested (8 µM) (*Figure 7A*). This behavior was in stark contrast to wild-type CtpS, whose activity plummeted to 4% of its maximum. Thus, at low enzyme concentrations, CtpS$^{E277R}$ exhibited slightly lower activity than wild type while at high enzyme concentrations CtpS$^{E277R}$ activity was significantly greater than that of wild type. One explanation for

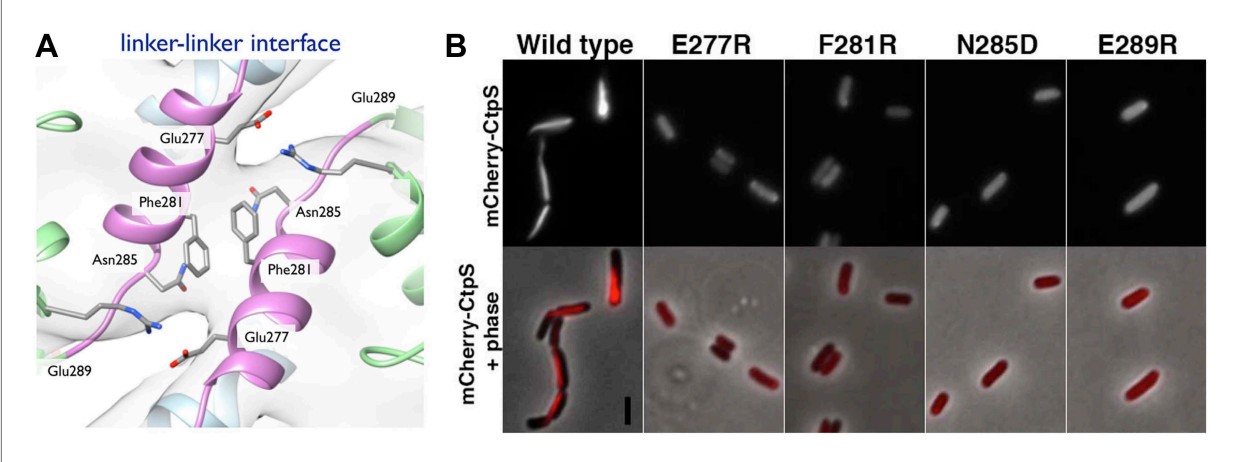

**Figure 6**. Linker helix residues form a polymerization interface. (**A**) The positions of the four polymerization mutants in the model of the linker–linker filament assembly interface. (**B**) Point mutants were engineered into an mCherry-CtpS fusion and imaged upon expression in *E. coli*. Scale bar = 3 microns. Wild type mCherry-CtpS forms filaments while mutant mCherry-CtpSs show diffuse localizations.
The following figure supplement is available for figure 6:

**Figure supplement 1**. Sequence alignment of several CtpS primary sequences.

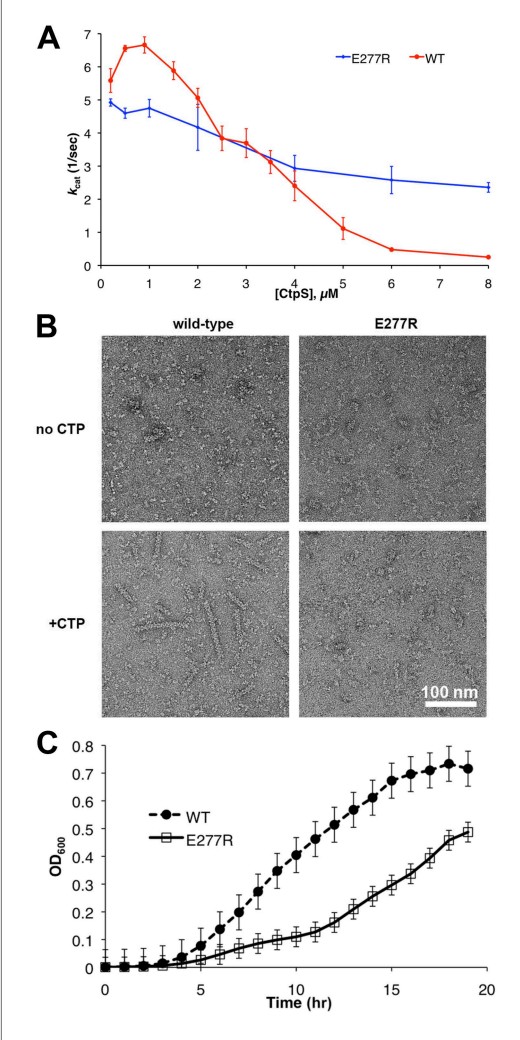

**Figure 7**. Linker helix mutations disrupt polymerization and cause a growth defect. (**A**) The CTP production activity of titrated levels of CtpS$^{E227R}$ exhibited a small decrease in enzymatic activity as enzyme concentration increases when compared to wild-type protein. Error bars = SD, n = 3–6. (**B**) Purified CtpS$^{E277R}$ does not polymerize in the presence of CTP. For both wild-type and E277R CtpS, there were very few filaments observed in the absence of CTP (top row). Upon the addition of nucleotide and MgCl$_2$, filaments were only observed in the wild-type sample (first column). (**C**) Growth curve comparing wild-type and CtpS$^{E277R}$ cells in LB media. CtpS$^{E277R}$ exhibits defective growth when compared to cells with wild-type CtpS. Both strains were grown overnight and subcultured into LB media. Growth curve comparing wild type to the defective growth of CtpS$^{E277R}$ mutant *E. coli* in minimal media. CtpS$^{E277R}$ mutants exhibit defective growth. Error bars = SE, n = 18.

The following figure supplements are available for figure 7:

*Figure 7. Continued on next page*

the comparatively modest decrease in CtpS$^{E277R}$ activity as a function of enzyme concentration is that CtpS$^{E277R}$ produces CTP, which at high CtpS concentrations can accumulate and competitively inhibit CtpS activity, resulting in a slight activity decrease. However, this mutant lacks the dramatic reduction in CtpS activity mediated by large-scale assembly into filaments. As predicted from thermodynamic linkage, the inability to polymerize also leads CtpS$^{E77R}$ to bind CTP less tightly, with a higher IC50 value than the wild-type enzyme (830 µM vs 360 µM at 200 nM enzyme, *Figure 7—figure supplement 2*). These data are thus consistent with the model that CtpS is negatively regulated in two ways: CTP competitively inhibits UTP binding, and large-scale assembly sterically hinders a conformational change required for CtpS activity. The quantitative differences between wild type and CtpS$^{E277R}$ activity suggest that large-scale assembly mediates rapid and efficient inhibition of enzymatic activity.

## The CtpS$^{E277R}$ polymerization interface mutant disrupts *E. coli* growth and metabolism

To determine the impact of CtpS$^{E277R}$ on cell physiology, we replaced wild-type CtpS with CtpS$^{E277R}$ at its native locus in *E. coli*. This strain exhibited defective growth compared to wild type in rich (*Figure 7C*) and minimal media (*Figure 7—figure supplement 3*). Wild type doubling time was 51 min ± 1.5 min, while the CtpS$^{E277R}$ doubling time was 130 min ± 11 min in rich media. Immunoblotting confirmed that CtpS$^{E277R}$ was expressed at similar levels to wild-type CtpS (*Figure 7—figure supplement 4*). One possible explanation for the growth impairment is that CtpS$^{E277R}$ could not produce enough CTP to support robust growth. However, CTP levels, as measured by mass spectrometry, are not reduced in the CtpS$^{E277R}$ strain (*Figure 8—figure supplement 1*). In fact, CTP levels are modestly higher in the mutant than in wild type cells (1.6 ± 0.3-fold higher). Because average CTP levels are higher in these cells, CtpS$^{E277R}$ likely does not impair growth due to reduced CTP production. Rather, the elevated CTP levels and the observation that growth became particularly affected at mid-log phase support the hypothesis that the CtpS$^{E277R}$ mutant is defective in regulating CTP levels when adapting to changes in the cellular environment.

Replacing wild-type CtpS with CtpS$^{E277R}$ also affected levels of other nucleotides and their precursors or byproducts (*Figure 8A*, *Figure*

*Figure 7. Continued*

**Figure supplement 1**. CtpS$^{E277R}$ does not polymerize in vitro.

**Figure supplement 2**. Polymerization enhances the inhibition of CtpS activity by CTP.

**Figure supplement 3**. Growth curve comparing wild type to the defective growth of CtpS$^{E277R}$ mutant *E. coli* in minimal media.

**Figure supplement 4**. CtpS protein levels are not depleted in the CtpS$^{E277R}$ mutant.

*8—figure supplement 1*). For example, the amount of the pyrimidine precursor orotate was $2.3 \pm 0.5$-fold reduced in the mutant, consistent with the idea that CtpS$^{E277R}$ is hyperactive and increases CTP production at the expense of its precursors. Together, these data indicate that disrupting the CtpS polymerization interface does not deplete CtpS or CTP. Instead, we hypothesize that CtpS$^{E277R}$ perturbs *E. coli* growth by disregulating nucleotide metabolism in a manner consistent with hyperactivating CtpS by disrupting a negative regulatory mechanism. These data are consistent with the observation that at the cellular concentration of CtpS, CtpS$^{E277R}$ is more active than the wild-type enzyme.

## CtpS$^{E277R}$ impairs negative feedback regulation in vivo

Steady-state measurements of metabolite levels cannot establish whether the observed increase in CTP levels corresponds to a defect in feedback inhibition of CtpS (as predicted by our model) or by stimulating CtpS activity in some other way. To directly assess feedback inhibition in vivo, we supplemented wild-type CtpS or CtpS$^{E277R}$ with C13-labeled cytidine, which is converted into C13-CTP by the nucleotide salvage pathway that functions independently of CtpS (*Ayengar et al., 1956*; *Valentin-Hansen, 1978*; *Fricke et al., 1995*). We note that nucleotide triphosphates cannot be imported into the cell such that we could not supplement with CTP itself. Furthermore, the use of C13-cytidine enabled us to use mass spectrometry to distinguish the CTP produced by nucleotide salvage (C13-CTP) from the CTP produced de novo by CtpS (C12-CTP). We hypothesized that if disruption of CtpS polymerization disrupts negative feedback, then CtpS$^{E277R}$ should maintain high CtpS activity despite the accumulation of C13-CTP from supplementation with C13-cytidine.

As predicted based on the independence of nucleoside import from nucleotide biosynthesis, the incorporation of C13-label into the CTP pool was similar in the wild type and CtpS$^{E277R}$ strains, indicating that both take up labeled cytidine and convert it into CTP at approximately the same rate (*Figure 8B*). In wild type cells, as the C13-CTP pool increased, the fraction of C12-CTP sharply decreased (*Figure 8C*). Thus, feedback regulation mechanisms compensate for the increased CTP production from cytidine by reducing de novo CTP production by CtpS. The decrease in the fraction of unlabeled CTP was less pronounced in the CtpS$^{E277R}$ mutant and by the end of the period assayed, unlabeled CTP levels were almost twofold higher in the CtpS$^{E377R}$ strain than in wild type (*Figure 8—figure supplement 2*). This result supports our conclusion that CtpS$^{E277R}$ hyperactivates CtpS by disrupting its negative feedback regulation and that this hyperactivation more than compensates for its reduced enzymatic activity. Since disruption of just one interaction in the proposed polymerization interface weakened the ability of CtpS to control CTP production even when all other forms of CtpS regulation are unaltered, we predict that any disruption of regions of inter-tetrameric contact, either by changes to the protein sequence or by chemical perturbation, would cause this deleterious regulatory defect.

## Coupling activity to polymerization enables ultrasensitive enzymatic regulation

What is the benefit of using polymerization as a negative-feedback regulation strategy? To quantitatively assess the impact of polymerization-mediated enzymatic inhibition, we developed a simple mathematical model of CtpS inhibition by CTP-dependent polymerization (see *Supplementary file 1* for details). A key point of the model is that the concentration of CtpS needed for polymerization depends on the free energy of polymerization, which in turn depends on the UTP and CTP concentrations. One mechanism for how CTP induces reversible polymerization is by CTP binding more favorably to the filament than to the free tetramer. This model leads to two predictions dictated by thermodynamic linkage: (1) CTP should be a more effective inhibitor at CtpS concentrations that favor polymer formation, and (2) the presence of CTP should enhance polymer formation and the reduction in CtpS specific activity ($k_{cat}$) as CtpS concentration increases. Indeed, at 4 μM CtpS, near the concentration at which CtpS $k_{cat}$ is one half of its maximum due to polymerization ([CtpS]$_{0.5}$, 3.3 μM, *Figure 7A*), the CTP IC50 value is

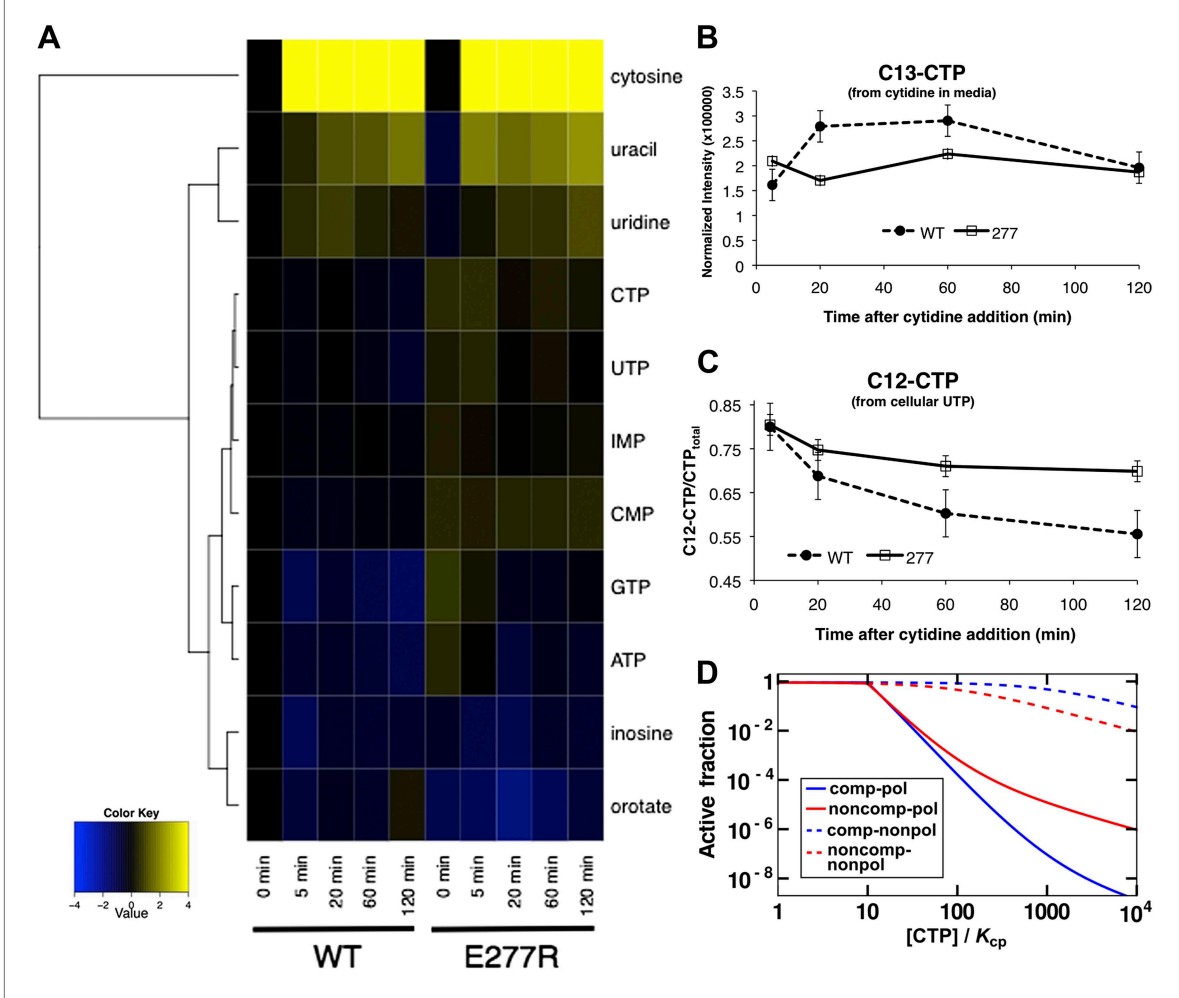

**Figure 8**. Mutation of polymerization interface disrupts CTP homeostasis in vivo. (**A**) Metabolic profiling of wild type and CtpS[E277R] mutant cells after addition of cytidine to minimal media. Nucleotide biosynthesis molecules are shown. (**B**) Incorporation of C13-label into CTP pool in wild-type and CtpS[E277R] mutant cells. Incorporation occurs at similar levels in both strains. Error bars = SE, n = 3. (**C**) The proportion of unlabeled (C12) CTP in wild-type and CtpS[E277R] mutant cells. The ratio of C12-CTP to total CTP is higher in the CtpS[E277R] strain. Error bars = SE, n = 3. (**D**) Model of the fraction of active (nonpolymerized and UTP-bound) CtpS, plotted vs CTP concentration. Comparison is shown between competitive inhibition with polymerization, noncompetitive inhibition with polymerization, competitive-nonpolymerizing, and noncompetitive-nonpolymerizing mechanisms. In all cases, we chose a fixed UTP concentration equal to $K_{cp}$, the dissociation constant of CTP and polymerized CtpS (see ***Supplementary file 1*** for details).

The following figure supplements are available for figure 8:

**Figure supplement 1**. Metabolomic analysis of wild-type and CtpS[E277R] *E. coli* after addition of 200 µg/ml C13-cytidine.

**Figure supplement 2**. CTP levels probed by mass spectrometry after addition of C13-labeled cytidine to the media.

**Figure supplement 3**. CTP binding enhances polymerization with a sharp response.

reduced to 170 µM, compared to 360 µM at 200 nM enzyme (***Figure 7—figure supplement 2***). Conversely, in the presence of 800 µM CTP, the [CtpS$_{0.5}$] value is 1.4 µM, reduced by more than half compared to that with no CTP (***Figure 7—figure supplement 3***). Interestingly, the presence of 400 µM CTP has only a small effect ([CtpS]$_{0.5}$ = 2.8 µM) , suggesting an ultrasensitive response of polymerization to CTP levels.

Another result of this polymerization-based mechanism is that the cooperativity of CTP-mediated inhibition increases as a function of the nucleation barrier to polymerization. Experimentally, the abundance of long polymers in vitro (***Figure 1B***) and the small number of polymers per cell in vivo (***Ingerson-Mahar et al., 2010***) suggest that CtpS polymerization exhibits a significant nucleation barrier. The

conformational differences between the free and filament forms of CtpS (*Figure 4*) may play a role in establishing this barrier. This barrier could result from the free energy change required to take the CtpS tetramer from a flexible 'free' state to more rigid 'filament' state upon the first assembly step of the polymer. Alternatively, dimerization of 'free' CtpS tetramers could allosterically influence one another to adopt the 'filament' conformation in a manner similar to one proposed for the cooperative polymerization of FtsZ (*Miraldi et al., 2008*). Our mathematical model enables us to estimate this nucleation barrier from the average polymer length, yielding a value of order 9 $k_BT$, where $k_BT$ is the thermal energy. Moreover, it demonstrates that coupling activity to polymerization with such a significant nucleation barrier represents a mechanism for generating extremely sharp transitions in enzyme activity.

We compared the sharpness of enzyme inhibition in our novel polymerization-based mechanism to that of previously characterized mechanisms of enzyme inhibition such as competitive and allosteric inhibition (*Figure 8D*; *Supplementary file 1*). We found that, among the mechanisms examined, the ones involving polymerization-based negative feedback yield the sharpest decrease in enzyme activity when CTP levels are increased, thereby enabling tight regulation of CTP production by CTP levels. Our estimate based on average CtpS filament length of the value of the nucleation energy yields extremely sharp transitions (see *Figure 8D*, where this estimate was used, and our discussion of response coefficients in *Supplementary file 1*). This sharpness is apparent in comparing the concentration dependences of CtpS specific activity in the presence of CTP. The $CTPS_{0.5}$ value at 400 µM CTP is slightly shifted compared to no CTP. At 800 µM, the $CtpS_{0.5}$ value is substantially decreased and the curvature more concave (*Figure 8—figure supplement 3*).

Because the onset of the decrease of activity can become arbitrarily sharp as the nucleation energy is increased, polymerization-mediated regulation is fundamentally different from the case of fixed stoichiometry enzyme oligomers, such as hemoglobin, that cooperatively bind an inhibitor. Another crucial difference with respect to such simple cooperative inhibition is that the polymerization-based mechanism also mediates negative feedback on CtpS activity from CtpS levels (*Supplementary file 1*). Hence, this mechanism uniquely enables ultrasensitive regulation of CtpS activity by both CTP and CtpS concentrations. Additionally, sequestering CtpS tetramers into the inactive filament ensures the availability of a CtpS pool that can be rapidly reactivated, limited only by the polymer disassembly rate. Our biochemical data confirm that depolymerization and subsequent repolymerization can occur within seconds (*Figure 1E*), while investigation of the in vivo kinetics of CtpS filament assembly and disassembly presents an interesting subject for future study.

## Discussion

Our studies suggest that in addition to being regulated by small-scale oligomerization, allosteric control, competitive inhibition, and transcriptional and post-translational mechanisms, CtpS is also regulated by large-scale assembly into filaments comprising hundreds of subunits (*Figure 1C*). CtpS polymerization is cooperative, which we conclude based on light scattering dynamics, the long polymers observed by EM, and the large fraction of polymerized protein observed by sedimentation (if assembly were non-cooperative one should always observe more tetramers than polymers). CtpS polymerization inhibits CtpS activity. The polymerization of CtpS is stimulated by binding its product, CTP, and disrupted by binding its substrates, UTP and ATP (*Figures 1E and 9*). Inter-tetramer interactions in the CtpS polymer sterically inhibit a conformational change that is thought to be necessary for CtpS activity, and mutations that disrupt polymerization disrupt CtpS regulation with significant impacts on cell growth and metabolism.

### The benefits of harnessing polymerization as a regulatory mechanism

With so many regulatory strategies in place, why add another? First, layering multiple levels of regulation results in robust regulatory control with a series of fail-safes that protects the cell from disregulated nucleotide levels. CtpS is a key node in nucleotide metabolism because it binds ATP, UTP, CTP, and GTP. We propose that strict regulation of nucleotide levels is so critical to controlled growth and division that CtpS evolved as a master switch to integrate information about nucleotide abundances and maintain their proper levels and proportions. Nucleotide biosynthesis is both energetically costly and controls the availability of raw materials for replication, transcription, and other biosynthetic pathways. Thus, coordinating biomass accumulation and cellular proliferation requires extremely tight control of nucleotide levels via CtpS that no one regulatory mechanism could achieve on its own. The need for such tight regulation could also explain recent observations that small CtpS polymers can

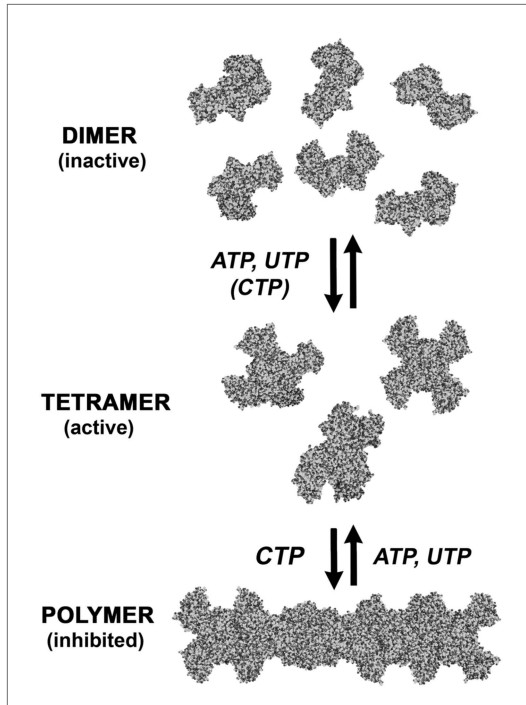

**Figure 9**. An expanded description of CtpS assembly. As shown in **Figure 1A**, tetramer formation from CtpS dimers is favored by a combination of enzyme concentration as well as nucleotide (substrates ATP and UTP or product CTP) and Mg$^{2+}$ binding. CTP binding and higher enzyme concentration further stimulates reversible formation of inhibited polymeric filaments, which can be disassembled by ATP/UTP.

combine to form higher-order larger structures (*Gou et al., 2014*) and can co-localize with other proteins involved in nucleotide metabolism (reviewed in *Carcamo et al., 2014*).

The second advantage of employing multiple types of regulation is that each regulatory strategy has distinct kinetics that together enables regulation over a wide range of potential conditions. For example, transcriptional regulation is slow in comparison to regulation by ligand binding. Competitive or allosteric regulation by ligand binding can be cooperative if the enzymes form oligomers, as in the case of hemoglobin (*Perutz, 1989*). However, the amassed activity of such oligomers is strictly linear with respect to protein concentration. By contrast, our modeling indicates that coupling activity to ligand-induced polymerization is a simple mechanism for promoting cooperativity with respect to protein concentration, while at the same time maintaining cooperativity with respect to ligand binding. An added benefit of polymerization-mediated inhibition is that it enables cells to sequester CtpS in an activity-primed tetramer state such that CtpS can be rapidly reactivated in a manner limited only by enzyme depolymerization (*Figure 9*). Previous models for enzyme sequestration have relied on the idea of preventing substrate binding (e.g., [*Jackson-Fisher et al., 1999*; *Michaelis and Gitai, 2010*]). Here, we propose an alternate mechanism for sequestration where the active sites can readily access substrates but conformational changes required for activity are restricted. While our data are consistent with the model of cooperative regulation by assembly, experimental noise and nonlinearities limit the current ability to measure the extent of that cooperativity, raising the possibility that there are yet more undiscovered features of CtpS regulation. As methods for manipulating and monitoring nucleotide levels become more available, it will also be interesting to determine the kinetics of the various CtpS regulatory mechanisms in vivo.

## Do other enzymes utilize polymerization-based regulation?

Though we have only tested the *E. coli* CtpS enzyme, we hypothesize that other prokaryotic and eukaryotic CtpS proteins may be subject to inhibition by polymerization. *Caulobacter crescentus* CtpS disassembles in the presence of DON while *Saccharomyces cerevisiae* CtpS shows longer filaments when cells were exposed to additional CTP (*Ingerson-Mahar et al., 2010*; *Noree et al., 2010*). The linker region implicated in *E. coli* CtpS polymerization is also mutated in three independent human lung carcinoma samples (*Forbes et al., 2008*), suggesting that metabolic regulation by CtpS polymerization is important for limiting human cell proliferation.

In the future, it will be interesting to determine if other enzymes employ polymerization-mediated regulatory strategies. In particular, we predict that enzymes that function at key metabolic nodes would most benefit from the ultrasensitive regulation provided by polymerization. Such cooperative assembly can coordinate the mobilization or sequestration of functional units, thereby dynamically altering the level of active enzyme without altering the overall enzyme concentration. The ultrasensitive kinetics of this transition would allow cells to rapidly respond to short-term changes in their environment or metabolic needs. For example, immediately following cell division, daughter cells could depolymerize any CtpS filaments inherited to compensate for reduced CtpS concentrations (perhaps from unequal partitioning) faster than translating and folding new proteins. The rapid kinetics of polymerization

could sequester CtpS when CTP is plentiful to prevent futile biosynthesis. A handful of other metabolic enzymes have been shown to form filamentous or large scale structures in vitro and in vivo (**Barry and Gitai, 2011**). CtpS may thus emerge as a model for a larger class of enzymes that are regulated by higher-order assembly to achieve cooperative enzyme activation or inactivation.

## Enzymatic regulation may have driven the evolution of large-scale polymers

Large-scale polymers such as cytoskeletal filaments play an essential role in organizing the cell. But how did such cytoskeletal polymers evolve? Our findings suggest that the selective benefit conferred by improving enzymatic regulation may have led to the evolution of large-scale filaments. Once present, these enzymatic polymers could then be appropriated for the structural functions commonly associated with the cytoskeleton. Finally, gene duplication and divergence would enable uncoupling and specialization of the enzymatic and structural properties of these proteins (**Barry and Gitai, 2011**).

The observation that CtpS polymerization is conserved among diverse prokaryotes and eukaryotes supports the hypothesis that CtpS polymerization arose in an early common ancestor and is a key feature of CtpS regulation. An example of appropriating an enzymatic polymer for structural functions comes from *C. crescentus*, where CtpS filaments regulate cell shape in a manner that can be uncoupled from their enzymatic activity (**Ingerson-Mahar et al., 2010**). While the enzymatic activity and polymerization capacity of CtpS is universally conserved, its cell shape function appears to be species specific. Thus, polymerization appears to have evolved early to regulate enzymatic activity while CtpS polymers were only later adapted for a structural role.

A similar evolutionary path could explain the structural similarity between hexokinase enzymes and the actin family of cytoskeletal elements (**Holm and Sander, 1993**; **van den Ent et al., 2001**). Specifically, we hypothesize that actin and hexokinase may have shared a common ancestor that, like CtpS, evolved polymerization as a regulatory mechanism. Gene duplication and divergence may have subsequently enabled actin to specialize as a structural element, while additional layers of enzymatic regulation may have obviated the need for hexokinase assembly (mammalian hexokinase does not polymerize). In this way, CtpS assembly and regulation may provide insight into the origins of the intracellular structural network that became the modern cytoskeleton.

# Materials and methods

## *E. coli* strains

| Strain | Description | Reference |
|---|---|---|
| ZG247 | NCM3722 | (*Soupene et al., 2003*) |
| ZG1075 | pyrG-His in BL21 * (DE3) | (*Ingerson-Mahar et al., 2010*) |
| ZG1076 | pyrG$^{E155K}$-His in BL21 * (DE3) | This study. |
| ZG1077 | pyrG$^{E277R}$-His in BL21 * (DE3) | This study. |
| ZG1082 | mCherry-CtpS in NCM3722 | (*Ingerson-Mahar et al., 2010*) |
| ZG1083 | mCherry-CtpS$^{E277R}$ in NCM3722 | This study. |
| ZG1084 | mCherry-CtpS$^{F281R}$ in NCM3722 | This study. |
| ZG1085 | mCherry-CtpS$^{N285D}$ in NCM3722 | This study. |
| ZG1086 | mCherry-CtpS$^{E289R}$ in NCM3722 | This study. |
| ZG1168 | CtpS$^{E277R}$-kan$^R$ chromosomal integrant in NCM3722 | This study. |
| ZG1169 | WT-kan$^R$ chromosomal integrant in NCM3722 | This study. |

## CtpS purification

Wild-type CtpS was purified as described previously (*Ingerson-Mahar et al., 2010*). CtpS-E155K and CtpS-E227R were purified as described previously with the exception that the 6XHis affinity tag was not cleaved in these cases. Similar treatment of the wild-type protein proved indistinguishable from the cleaved sample.

## Activity/polymerization assay

Purified CtpS protein was incubated at 37°C for 20 min in 50 mM Tris HCl (pH 7.8), 10 mM MgCl$_2$, 1 mM UTP, 1 mM ATP, and 0.2 mM GTP to allow tetramer formation. CTP production was initiated by the addition of 10 mM glutamine to create a full activity buffer (referred to in text at 'activity buffer') (*Ingerson-Mahar et al., 2010*) immediately prior to recording of sample measurements. Time between glutamine addition and initiation of sample recording averaged 5 s was based on the amount of time required to load the sample. Reaction was monitored at 37°C for 5 min in a QuantaMaster 40 Fluorometer (Photon Technology International, Birmingham, NJ) equipped with photo multiplier tubes for both scattering and transmittance. Right angle light scattering at 405 nm with a 1 mM slit width detected polymerization, and transmittance at 291 nm with a 0.25-mM slit width detected CTP production with both values reported in arbitrary units. Reactions were performed in 150 µl samples. Polymerization was monitored for 3 min unless otherwise noted. Detection of light scattering and transmittance alternated with an integration time of 1 s. CTP production velocity ($k_{cat}$, µmol/s) was determined for the first 30 s of the reaction. CTP production was normalized by the concentration of CtpS enzyme in each sample. Due to the fluorometer assay's use of transmittance and a photon multiplier, we compared data collected to data collected over the same concentration range on a more traditional spectrophotometer setup in the Baldwin lab. Comparison yielded the presence of a scaling factor to be applied to the fluorometer data set to yield $k_{cat}$ ranges consistent with published data. Data were scaled to yield the same maximal $k_{cat}$ value for both data sets. The fold-change in activity over the concentrations was similar between the data sets. Overlay of the data are shown in *Figure 1—figure supplement 7*. Quantification of polymerization was calculated using the difference between the average initial and final values of light scattering for each sample (n = 5 for average) in *Figures 1B and 2A* and *Figure 1—figure supplement 1*, *Figure 2—figure supplements 1 and 2*, and *Figure 7—figure supplement 1*. All other light scattering values are the actual values of light scattering recorded (in arbitrary units), except where noted in the figure legends.

## CTP production activity assay

Enzyme concentration was determined using the extinction coefficient for CtpS, 0.055 µM/A$_{280}$ unit. Concentrated enzyme (40–80 µM) was annealed at room temperature for 3 min at 21°C in 10 mM MgCl$_2$, 60 mM HEPES pH 8.0, then mixed with 1.5 mM ATP and 600 µM UTP and incubated for 20 min at 37°C. 4-min incubations with substrates gave equivalent results. When CTP was present, it was included in the ATP/UTP mixture. The reactions were initiated by mixing with 10 mM final glutamine and the absorbance at 291 nm measured. It was not possible to measure the rates of 277R above 8000 nM (19 µM/s) because the rate could not be reliably measured considering the dead time of the instrument and the procedure (~5 s). The final reactions contain 0.1–25 mM NaCl from the enzyme storage stocks, but these concentrations of NaCl do not have noticeable effects on enzyme rate. The annealing step is critical for the highest specific activities from stocks stored frozen or at 4°C and is optimal at concentrations greater than 2 µM. From CTP inhibition experiments, the CTP IC50 value at 200 nM CtpS$^{WT}$, 600 µM UTP, and 1.5 mM ATP was 360 µM (*Figure 7—figure supplement 2*). The concentration-dependences were complex and yielded curved Hill plots. IC50 values were obtained by linear extrapolation using points flanking $v_i = 1/2v_o$. Graphical data points represent the averaged values of 2–6 experiments with error bars indicating the standard error or standard deviation of each measurement.

## CTP polymerization assay

Purified CtpS protein was incubated at 37°C for 20 min in 50 mM Tris HCl (pH 7.8) and 10 mM MgCl$_2$. 1 mM CTP (Epicentre. Madison, WI) was added immediately before the sample was loaded into the fluorometer. Time between CTP addition and initiation of sample recording averaged 5 s. Measurements were taken as described for the activity/polymerization assay.

## Ultracentrifugation activity assay

Purified CtpS protein was incubated in the activity buffer or CTP buffer (1 mM CTP, 10 mM MgCl$_2$, 50 mM Tris–HCl [pH 7.8]) at 37°C for 1 hr. Samples were centrifuged at 116,000×$g$ for 15 min at 4°C using an Optima TLA 100 rotor (Beckman, Indianapolis, IN). After centrifugation, the supernatant was removed. For activity assays, the pellet was resuspended in 100 µl ice cold buffer containing 50 mM Tris HCL (pH 7.8) and 10 mM MgCl$_2$. 10 µl of this CtpS pellet solution was added to complete activity buffer containing 50 mM Tris HCl (pH 7.8), 10 mM MgCl$_2$, 1 mM UTP, 1 mM ATP, 0.2 mM GTP, and 10 mM glutamine to monitor initial activity.

## Quantification of native CtpS levels

Wild-type NCM3722 was grown to early exponential phase in M9 minimal media plus 0.04% glucose (M9G). Native levels of CtpS were quantified based on a standard curve of purified CtpS and normalized based on the $OD_{600}$ of the culture. Calculations assume 1 OD unit = $8 \times 10^8$ cells and cellular volume = 1 µm³. Samples were loaded on a 10% Tris-glycine SDS PAGE gel. Membrane was probed with 1:15,000 rabbit anti-CtpS. Band intensities were compared using ImageJ.

## Quantification of CtpS in CTP buffer

For quantification of CtpS pelleting in variable CTP, 130 µg CtpS was incubated in 500 µl appropriate concentrations of CTP buffer (4.3 µM CtpS). 200 µl samples were spun at 116,000×*g* on a Beckman TLA-100 rotor for 30 min at 4°C. The pellet fraction was resuspended in 50 µl SDS-PAGE sample buffer. Samples were loaded on a 10% Tris-glycine SDS PAGE gel. Membrane was probed with 1:15,000 rabbit anti-CtpS. Band intensities were compared using ImageJ.

## Electron microscopy

### Negative stain imaging

Negative stain EM samples were prepared by applying polymerized CtpS to carbon-coated grids and staining with 0.75% uranyl formate (*Ohi et al., 2004*). 15 µM purified CTPs in 50 mM Tris HCl (pH 7.8) was incubated for 20 min at 37°C with 1 mM CTP and 5 mM $MgCl_2$, or without nucleotide as a control. Reactions were diluted 1/10 in the same buffer supplemented with 50% glycerol before being coated onto grids and stained with uranyl formate for analysis. Protein purifications for wild-type CTPs and mutants E155K and E277R were performed simultaneously. Negative stain EM was performed on a Tecnai TF20 microscope (FEI Co.) operating at 200 kV, and images were acquired on a 4 k × 4 k CCD camera (Gatan, Inc.). Micrographs all taken at 55,000X magnification.

### Cryo-EM imaging

15 µM purified CTPs was incubated for 20 min at 37°C in activity buffer. Samples were prepared by applying polymerized CtpS to glow-discharged Quantifoil holey-carbon grids (Quantifoil Micro Tools GmbH, Jena, Germany), blotting in a Vitrobot (FEI Co., Hillsboro, OR), and rapidly plunging into liquid ethane. Cryo-EM data were obtained on a Titan Krios operating at 200 kV with a 4 k × 4 k Gatan Ultrascan camera at a pixel size of 0.82 Å/pixel. Total electron dose was in the range of 25–30 e−/Å² per image, and images were acquired over a defocus range of −1 to −3.5 µm (average −2.5 µm).

## Image processing

Defocus parameters for each micrograph were determined with CTFFIND (*Mindell and Grigorieff, 2003*). CTF correction was achieved by applying a Wiener filter to the entire micrograph. Lengths of helix were defined in the boxer program of the EMAN software suite (*Ludtke et al., 1999*). Overlapping segments were extracted from the CTF-corrected micrographs along the length of each helix. In total, 12,465 overlapping segments were extracted in 510 × 510 Å boxes, representing approximately 56,000 unique CtpS monomers. Segments were binned twofold prior to reconstruction, at a final pixel size of 1.64 Å. Iterative helical real space reconstruction (IHRSR) was performed essentially as described by *Egelman (2007)* and *Sachse et al. (2007)*, using SPIDER (*Frank, 1996*) for projection matching and back projection, and hsearch_lorentz (*Egelman, 2000*) for refinement of helical symmetry parameters. A cylinder was used as the initial reference volume, and 30 rounds of iterative refinement were carried out at increasingly smaller angular increments (1.5° in the final round). A preliminary reconstruction was performed imposing only helical symmetry, from which it was clear that the repeating helical sub-unit was the CtpS tetramer; in subsequent runs of IHRSR the local 2-2-2 point group symmetry of the CtpS tetramer was also enforced. Visualization of the cryo-EM reconstructions and rigid body fitting of the CtpS crystal structure into the EM map were performed in Chimera (*Pettersen et al., 2004*). The CtpS crystal structure monomer was initially fit as a single rigid body into the EM map, followed by local refinement of the fit treating the two domains and linker region as three separate rigid bodies. The final EM map was amplitude corrected using amplitudes from the atomic model.

## Site-directed mutagenesis

Site-directed mutagenesis was performed using the QuickChange (Agilent, Santa Clara, CA) system with minor modifications to enable using KOD polymerase (Millipore, Billerica, MA) or GXL polymerase (Takara, Mountain View, CA).

### Live cell imaging

Strains were grown overnight in LB with 50 µg/ml carbenicillin, subcultured, and grown until early exponential phase. Fluorescent protein expression was induced with 0.01 mM IPTG for 2–3 hr. Cells were immobilized on 1% agarose in water pads containing 0.01 mM IPTG. Imaging was performed using a Nikon (Melville, NY) TI-E microscope using a 100X Nikon Plan Apo objective (NA = 1.4), Chroma ET572/35X (excitation) and ET622/60M (emission), Prior Lumen 200 Pro illumination, and 89014VS dichroic mirro. Images were acquired with an Andor Clara camera using NIS-Elements software.

### Chromosomal integration of CtpS$^{E277R}$

PCR fragments of the region from *mazG* to *ygcG* either containing a wild-type *pyrG* or *pyrG*$^{E277R}$ coding region and a kanamycin resistance cassette between *eno* and *ygcG* were integrated into the NCM3722 chromosome by Lamda red recombination. Recombineered cells were recovered on LB agar with 50 µg/ml kanamycin and 200 µg/ml cytidine.

### Growth curves

Strains were grown overnight in LB containing 30 µg/ml kanamycin and 200 µg/ml cytidine. Then cells were diluted to the same OD in 100 µl LB plus kanamycin or M9G plus kanamycin (as noted) in a 96-well format. $OD_{600}$ was recorded using a BioTek (Winooski, VT) microplate reader at 37°C with continuous shaking.

### Metabolomics of CtpS$^{E277R}$ chromosomal integrant

Strains were grown in M9 minimal media to early exponential phase. Media were supplemented with 13C5-ribose-labeled cytidine (Cambridge Isotopes, Tewksbury, MA) to a final concentration of 200 µg/ml, and cell growth was continued at 37°C. Sample preparations were modified based on *Lu et al. (2007)*. Specifically, 24 milliliters of bacterial cultures were harvested by centrifugation at room temperature at five time points following cytidine addition: 0 min, 5 min, 20 min, 60 min, and 120 min. The pellet was resuspended in 1 ml 40:40:20 methanol:acetonitrile:water quenching buffer and allowed to sit on dry ice for 15 min. Sample was spun at maximum speed in a microcentrifuge for 5 min at 4°C. Then the resulting pellet was resuspended again in 0.6 ml fresh 40:40:20 solution for 15 min on dry ice and then spun as before to quench and extract metabolites a second time. Quenching buffer supernatants were combined and concentrated threefold for mass spectrometry as in *Xu et al. (2012)*.

### Accession numbers

The cryo-EM map of the CtpS filament has been deposited with the Electron Microscopy Data Bank [EMDB] accession number EMD-2700.

## Acknowledgements

We acknowledge the members of the Gitai lab, in particular E Klein, A Siryaporn, R Morgenstein, and M Wilson, for helpful discussions and N Ouzounov for assistance in strain construction. In addition to the Rabinowitz lab, we specifically thank W Lu and J Rabinowitz for advice and J Fan for assistance with metabolic profiling. Crl antibodies were a gift of the Silhavy lab. We are grateful to the Facility for Electron Microscopy Research at McGill University for use of electron microscopes and for staff assistance. We would like to thank M Shepherd for assistance in image processing.

## Additional information

#### Funding

| Funder | Grant reference number | Author |
|---|---|---|
| Human Frontier Science Program | | Anne-Florence Bitbol, Ned S Wingreen, Justin M Kollman, Zemer Gitai |
| National Institutes of Health | 5RO1GM107384 | Zemer Gitai |
| National Science Foundation | PHY-0957573 (partial support) | Anne-Florence Bitbol |

| Funder | Grant reference number | Author |
|---|---|---|
| National Institutes of Health | R01GM082938 (partial support) | Anne-Florence Bitbol |

The funders had no role in study design, data collection and interpretation, or the decision to submit the work for publication.

## Author contributions

RMB, A-FB, JMK, Conception and design, Acquisition of data, Analysis and interpretation of data, Drafting or revising the article; AL, EJC, CHH, JMH, H-JL, Acquisition of data, Analysis and interpretation of data; EPB, NSW, ZG, Conception and design, Analysis and interpretation of data, Drafting or revising the article

## Additional files

### Supplementary file

• Supplementary file 1. Model of CtpS polymerization and inhibition.

### Major dataset

The following dataset was generated:

| Author(s) | Year | Dataset title | Dataset ID and/or URL | Database, license, and accessibility information |
|---|---|---|---|---|
| Kollman JM, Charles EJ, Hansen JM | 2014 | Cryo-EM structure of the CTP synthetase filament | http://www.ebi.ac.uk/pdbe/entry/EMD-2700 | Publicly available from The Electron Microscopy Data Bank (EMDB). |

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
