## [Decision Letter]

Thank you for resubmitting your work entitled “Large-scale filament formation inhibits the activity of CTP synthetase” for further consideration at *eLife*. Your revised article has been favorably evaluated by Michael Marletta (Senior editor), a member of the Board of Reviewing Editors (Mohan Balasubramanian), and one of the original reviewers. Reviewer 1 agrees the paper is much improved, but has raised some issues.

We will be pleased to accept the paper for publication in *eLife*, provided you can consider the points raised by Reviewer 1 and address the key points. We believe additional experiments are not required, but please consider rewriting the paper to more precisely state your case. In particular, please consider the point pertaining to cooperative assembly and assembly threshold as mentioned by the reviewer in his/her first point. I hope to receive a response letter and a revised submission soon.

*Reviewer 1*:

This paper has been improved by many clarifications and by some new data. The clarifications have in some cases revealed new concepts, but also new problems and need for additional clarification. I still feel that the case for an assembly threshold and therefore cooperative assembly is weak. But if this can be clearly addressed, and additional points clarified, I recommend publication. I think this study will have an initial impact and will be a good starting point for more comprehensive study of cooperative assembly. Detailed comments follow.

In the previous review, I suggested that the light scattering curve did not support the conclusion that there was a critical concentration for assembly. The authors have moderated their conclusion and avoided the term critical concentration. They now state that ”CtpS has a threshold concentration of 1-2 μM (Figure 1—figure supplement 1), above which an increase in right-angle light scattering indicates assembly into polymers”. However, I fail to see how the LS curve in Figure 1 can be fit with a threshold concentration different from zero. Figure 2 seems a bit better, but I doubt that a straight line fit would indicate a threshold with any statistical significance. The log-log curves in Figure 1—figure supplement 1 (which would hit zero at about 0.3 µM, not 1-2 µM) are also not convincing, especially since values below zero were switched to positive. I believe the important signals at low concentration are just lost in the noise. There is a much stronger case for a threshold of enzyme activity. Both Figure 1 and Figure 1—figure supplement 1 A make a reasonably convincing case that enzyme activity drops only after a 1-2 µM. I think it would be better to base the idea of a threshold on enzyme activity, and admit that this threshold for assembly is obscured by the noise in the LS curve.

“Upon addition of these substrate nucleotides, we observed a sharp decrease in light scattering that corresponded to a sharp increase in CtpS activity. This transition was followed by a gradual increase in light scattering and corresponding decrease in activity back to initial residual level (Figure 1).” I agree that the decrease in LS is indeed sharp, but the increase in activity is much slower. Also, although the LS indicates reassembly, there is no “corresponding decrease in activity”. It actually remains at its peak value or shows a small steady increase. This contradicts the story, but I think the interpretation should be changed to note this discrepancy.

“The cellular level of CtpS protein in E. coli grown in minimal media was 2.3 μM (Figure 1—figure supplement 2), indicating that the CtpS polymerization observed in vitro occurs within a physiologically relevant concentration regime.” This is actually right at the borderline of where in vitro activity starts to decrease. I would conclude that the proposed regulation mechanism by assembly is on the borderline of physiological significance.

In Figure 2 I think I understand that this is in buffer containing no substrates. Please clarify this in the legend. For Figure 2 please give the concentration of CtpS. I note in that the maximum CtpS in the pellet is about 50%. Does that fit with the predicted threshold concentration? It would if the CtpS were 4 µM, but not if it were 10 µM.

“activity buffer containing saturating amounts of substrates (UTP, ATP, and glutamine) as well as GTP and Mg2+ (referred to as “activity buffer” throughout the text”. But later ”by first allowing CtpS to polymerize in activity buffer and then adding 1 mM UTP and ATP.” Does activity buffer contain UTP/ATP?

“CtpS's product, CTP, is both necessary and sufficient to induce CtpS polymerization.” There is no CTP in activity buffer, but polymerization occurs, Figure 1. Ah, I see explained in the response to the original review that the assembly induced in Figure 1 is now thought to be produced entirely by the CTP generated by the CtpS when all substrates are present. Wow. The authors should be aware that I completely missed this in the text. I would suggest warning the reader after Figure 1 that, although it looks like the substrates are inducing assembly, it is actually the product CTP, which is produced only when all substrates are present, that is causing assembly. And then when discussing Figure 2 (and Figure 2—figure supplement 1) the interpretation should be emphasized again and all evidence summarized.

This revelation now presents additional questions for Figure 1. Methods state that LS was monitored for 5 minutes, but later “for 3 minutes unless otherwise noted.” But what was the level of CTP generated in this time? Figure 2 shows that 1 mM CTP was needed for maximum assembly (of the unknown concentration of CtpS). Since the substrates were present at 1 mM, this would be the maximum amount possible CTP that could be generated. The question is, how fast does it approach the 0.5-1.0 mM level? This would seem to introduce an additional level of complexity to the kinetics, and it would seem important to see a couple of curves showing the kinetics of CTP production, say at high and low CtpS concentrations. Most important would be to know what level of CTP was produced at the 3 minute time point? Again we would need this for a high and low CtpS. Best to show some kinetic curves in SI.

This also suggests that the attempt to identify a polymerization threshold in 1B may be complicated by the kinetics of CTP formation. It would seem better to shift the search for a polymerization threshold to the conditions in 2A, a clean system with constant CTP. Unfortunately, the LS system has already shown itself inadequate. But how about the centrifugation assay, used only once in 2B. Might this be a better assay to indicate a polymerization threshold.

“GTPase assembly is repressed by its substrates” This is also not clear to me. Maybe it is from Figure 2, but that is complicated by the DON inhibitor.

The model presented in SI proposes a dimer nucleation step, which is difficult to imagine for a linear polymer where all subunit interfaces are the same. This enigma has been discussed for FtsZ, which is also one subunit thick, and for which cooperative assembly is convincingly established. A possible resolution has been addressed by Miraldi and Romberg Biophys J 95:2470, which the authors should consider.

[Editors’ note: a previous version of this study was rejected after peer review, but the authors submitted for reconsideration. The previous decision letter after peer review is shown below.]

Thank you for choosing to send your work entitled “Large-scale filament formation inhibits the activity of CTP synthetase” for consideration at *eLife*. Your full submission has been evaluated by Michael Marletta (Senior editor), a member of our Board of Reviewing Editors, and 3 expert peer reviewers, and the decision was reached after discussions between the reviewers. We regret to inform you that your work will not be considered further for publication at this stage.

The following individuals responsible for the peer review of your submission have agreed to reveal their identity (peer reviewer 2: Ji-Long Liu).

As you will see from the referees' comments, there is a great deal of enthusiasm for your study. However, referee 1 raises several substantive and important points, which the other referees concurred with in subsequent discussions. We believe the requisite revisions that address the key points raised by referee 1 (on the relationship between polymer formation and enzymatic activity) may not be accomplished in a 2 month period, within which we expect revisions to be submitted typically.

However, if you believe you can address all the issues raised, we will be happy to consider a suitably revised manuscript as a new submission that will be sent back to the same referees. In this instance, we would encourage you to fully address the points.

*Reviewer 1*:

This paper studies the polymerization of the enzyme CtpS into filaments, and provides some evidence that the polymerization inhibits enzyme activity, and may thus be a regulatory mechanism. A cryoEM reconstruction of the filaments shows a good fit to existing crystal structures, but negates the obvious possibility that the polymers blocked activity by occluding active sites – they are all exposed in the polymer. The authors are left with the speculation that the polymer may lock the subunits into a certain conformation, and block its switch to another conformation needed for activity. Unfortunately there is no information on whether there really is another conformation. Overall the cryoEM is probably well done, even if it doesn't lead to more than a speculation on a mechanism.

The structure is accompanied by experimental measures of polymerization, attempting characterize the assembly mechanism and to link it to inhibition of enzyme activity. I find these to be far too preliminary, and with confusing and erroneous interpretations. I therefore cannot recommend publication. And I can't recommend simple fixes and revisions. I think these studies need to be greatly expanded, especially with kinetics.

Detailed comments:

1) The “activity buffer” apparently came from an earlier paper measuring the enzyme activity. Curiously it contains no monovalent cations. I think it would be important to provide at least a quick test of how assembly and enzyme activity are affected by 300 mM potassium, i.e., in a physiological buffer.

2) The existence of a critical concentration would imply a cooperative assembly, and this is an important issue. But I don't think the curve in Figure 1 is convincing. A Cc should be seen as a flat line of zero assembly below Cc, and a straight line of constant slope above. The curve in 1a might be described as a small slope from 0-7 µM, a steeper rise to 10, and approaching a plateau by 20. However the points are erratic and might also be described as a single line from 0-20. I see error bars on only two points. I think it is impossible to judge a mechanism from this erratic LS data, and certainly a Cc is not convincing. The logarithmic plots in S1 don't help, since a Cc is best seen in a linear plot. The Figure caption is sloppy. It indicates a log plot in an inset (but it is in SI), and 1c is confusing. It is concluded that activity increases as the polymers in the pellet disassemble, but the disassembly is not confirmed by any measurement.

3) “polymers are inactive or significantly less active.” If there were a Cc, and polymers were inactive, one would expect to see activity increase in direct proportion to CtpS concentration below Cc, and flat-line above Cc. The mechanism is clearly more complex. “CtpS was incubated in activity or CTP buffer [this is not defined] for one hr...centrifuged for 15-30 minutes.” A science paper should report the exact conditions used for the data shown.

4) Polymerization was apparently initiated by adding glutamine, and the monitored for 3 minutes. The initiation by glutamine was referenced to their 2010 paper, but that paper showed no kinetic analysis. We really need a detailed analysis how glutamine induces polymerization - a time course of LS and enzyme activity, as a function of CtpS concentration. This is likely to invite a more complex analysis, including the question of nucleation, but that should be a part of any conclusion about a Cc. At the very least we need to know that the LS and enzyme activity plotted are at steady state. Figure 2 suggest that LS is still increasing at 3-5 minutes.

5) The negative stain is poorly described: ” in 50% glycerol [no buffer?]...with or without CTP and MgCl2 [don't you need to state which were used for which figures?]...diluted 1/10 in storage buffer [not defined, and why was everything not in the same buffer as the assays?].

6) Figure 1 “sharp decrease in LS”: This may have been caused by agitation of the polymers as the substrate [what was the substrate, there are 3??] was added and mixed. This needs controls of buffer only.

7) Figure 2 has many of the same serious problems as Figure 1. It raises the observation that polymers can be initiated by glutamine, a substrate, or by CTP, the product, which I thought inhibits in presence of glutamine? This (and the inhibition by ATP and UTP) suggests that we really need a comprehensive study of how the substrates and CTP stimulate and inhibit assemble.

8) “…indicating that tetramer formation precedes filament formation.” This throws another major variable into the story. When one starts with CtpS in whatever buffer, is it a monomer or tetramer? Do substrates cause tetramer formation? CTP? If tetramers need to form before filaments, this will greatly complicate the assembly mechanism.

9) The cryoEM can't really be judged by non-experts, but a couple of things caught my attention. It is first stated that the filament is fit by x-shaped CtpS tetramers. Then in 4b the structure is fit by CtpS monomers, with a (apparently flexible) linker between the two domains. What happened to the tetramers, and how do the monomer domains in the final fitting compare to their position in the tetramers? OK, I see in 6A that the fitted (flexible) monomers seem to superimpose almost exactly on the tetramer. I don't understand the conclusion ”a compression of the tetramer along the filament axis [not defined, but maybe vertical on the figure?]. The superposition seems to me extremely precise.

10) “There is strong density for CTP at the inhibitory site.” Is it really possible for a single CTP to produce strong density in a 10 Å cryoEM reconstruction? Even more so for the phosphates of ADP, and the missing density of the base - how could the base be partially disordered? The conditions for making the polymers for cryoEM are not given, but it seems CTP was used.

11) Figure 8 shows the speculative identification of 4 aa's that are on the linker and that might be part of a subunit interface. These were mutated, and all of the destroyed the in vivo localization – the mCherry is diffuse in the cytoplasm. However, the mCherry seems to be at the N terminus, and it could fold even if the fused CtpS were defective. Figure 9 provides some support for the suggestion that E277R does block polymerization and is accompanied by increased enzyme activity at the higher CtpS concentrations, where wt protein apparently loses activity upon polymer formation. If this were coupled with a more comprehensive study of kinetics of assembly, it would be an interesting if not definitive finding.

Reviewer 2

With a combination of biochemistry, electron microscopy, metabolomics and mathematical modelling, Gitai and colleagues determine the function and mechanism of CTP synthetase (CtpS) polymerization. They show that CtpS activity is negatively regulated by CtpS polymerization which is induced by excess CTP. By solving the structure of *in-vitro* synthesized CtpS polymers, the authors lighten the implications of the structure of these filaments on CtpS activity. When CtpS is polymerized the residues required for catalytic activity or conformational changes are not accessible therefore suggests that the CtpS should be free of filaments to be active. Finally the authors suggest that polymerization of CtpS has significance in terms of super-fast regulation of enzyme activity and therefore why these structures have been conserved through evolution.

This is an elegant study. The manuscript is one of the best manuscripts that I have reviewed in many months. The data are convincing and the authors have not made any major assertions, which their results do not seem to back up.

*Reviewer 3*:

The manuscript ”Large-scale filament formation inhibits the activity of CTP synthetase” by Barry et al is a fascinating study of the coupled polymerization and activity of the conserved enzyme CTPs. The authors use light-scattering in vitro experiments to show that CTPs's product, CTP, induces polymerization, and that polymerization inhibits activity. This direct connection was demonstrated by an elegant combination of structural work (TEM and cryoEM) and steady-state and dynamic in vitro polymerization studies. The authors then perform structure-guided mutagenesis to disrupt polymerization in vivo and demonstrate for one of these mutants that it is the inhibition that is removed (the mutant is not less active in CTP production). Finally, they show that this mutant has a severe growth defect, which is consistent with disregulation of CTP levels, rather a simple decrease in the CTP pool. To integrate these results within a framework, they develop a mathematical model that is parameterized quantitatively on their in vitro polymer length measurements, that demonstrates that the coupling they have uncovered allows for an ultrasensitive response to changes in environment.

This paper is comprised of excellent work, and sets a high standard for work studying the regulation of polymers of metabolic enzymes, a rapidly growing area in the last few years. There is an impressive correspondence between *the* in vitro and in vivo results, leading me to think that they have provided an elegant understanding of a complex system. Moreover, the central importance of CTP for physiology in all organisms makes the relationships they have revealed of critical importance for a broad set of fields including biochemistry, cellular physiology, evolutionary biology, and cell biology. I have only a few minor clarifications that would help (for me at least) readability.

1) Are there error bars in all plots that are in some cases invisible? If not, would be good to add, if so, would be good to change from the squares and diamonds to small points so that error bars are more visible.

2) Is it the case that the CTPs mutant cells in Figure 8 are smaller? They appear so to me; it would be useful to quantify this, and if this holds up, this is further evidence that the mutants are impacted in growth.

3) I was confused as to why they predicted that a multiple site mutant would have exacerbated effects; might this not be true if polymerization is already abolished in E277R?

---

## [Author Response]

*In the previous review, I suggested that the light scattering curve did not support the conclusion that there was a critical concentration for assembly. The authors have moderated their conclusion and avoided the term critical concentration. They now state that “CtpS has a threshold concentration of 1-2 μM (*Figure 1—figure supplement 1*), above which an increase in right-angle light scattering indicates assembly into polymers”. However, I fail to see how the LS curve in*
Figure 1
*can be fit with a threshold concentration different from zero.*
Figure 2
*seems a bit better, but I doubt that a straight line fit would indicate a threshold with any statistical significance. The log-log curves in*
Figure 1—figure supplement 1
*(which would hit zero at about 0.3 µM, not 1-2 µM) are also not convincing, especially since values below zero were switched to positive. I believe the important signals at low concentration are just lost in the noise. There is a much stronger case for a threshold of enzyme activity. Both*
Figure 1
*and*
Figure 1—figure supplement 1
*make a reasonably convincing case that enzyme activity drops only after a 1-2 µM. I think it would be better to base the idea of a threshold on enzyme activity, and admit that this threshold for assembly is obscured by the noise in the LS curve*.

We thank the reviewer for pointing out that the noise within the light scattering data makes the evaluation of a threshold concentration or critical concentration for polymerization difficult. We have rewritten this section to acknowledge that the light scattering and CTP production data must be taken together to support a potential critical concentration, the value of which is not easily derived but can be approximated.

*“Upon addition of these substrate nucleotides, we observed a sharp decrease in light scattering that corresponded to a sharp increase in CtpS activity. This transition was followed by a gradual increase in light scattering and corresponding decrease in activity back to initial residual level (*Figure 1*).” I agree that the decrease in LS is indeed sharp, but the increase in activity is much slower. Also, although the LS indicates reassembly, there is no “corresponding decrease in activity”. It actually remains at its peak value or shows a small steady increase. This contradicts the story, but I think the interpretation should be changed to note this discrepancy*.

CTP productions slows and then levels off to essentially no change in transmittance at 291 nm (the readout of CTP production), indicating that CTP is no longer accumulating at a detectable level.

*”The cellular level of CtpS protein in E. coli grown in minimal media was 2.3 μM (*Figure 1—figure supplement 2*), indicating that the CtpS polymerization observed* in vitro *occurs within a physiologically relevant concentration regime.” This is actually right at the borderline of where* in vitro *activity starts to decrease. I would conclude that the proposed regulation mechanism by assembly is on the borderline of physiological significance*.

We have moderated this statement to say: “…may be physiologically favourable within the cell”. In addition, in the Modeling section, we discuss how both the absolute amount of CtpS and the relative levels of substrate and product influence the nucleation barrier for polymerization, so the threshold for polymerization may be different in vivo.

*In*
Figure 2
*I think I understand that this is in buffer containing no substrates. Please clarify this in the legend. For*
Figure 2
*please give the concentration of CtpS. I note in*
Figure 2—figure supplement 3
*that the maximum CtpS in the pellet is about 50%. Does that fit with the predicted threshold concentration? It would if the CtpS were 4 µM, but not if it were 10 µM*.

We thank the reviewer for pointing out that the figure legends and Methods section for these experiments are ambiguous. The legends for Figure 2 have been updated to highlight the point that substrates are not present. The Methods section has been expanded to describe the assay in clearer detail, explaining that the [CtpS] present is 4.3 μM. As the reviewer points out, this should predict that some, but not all CtpS in the sample would be polymerized.

*“activity buffer containing saturating amounts of substrates (UTP, ATP, and glutamine) as well as GTP and Mg2+ (referred to as “activity buffer” throughout the text”*. *But later “by first allowing CtpS to polymerize in activity buffer and then adding 1 mM UTP and ATP.” Does activity buffer contain UTP/ATP?*

The reviewer is correct; as stated in the Abstract and in the Methods section, activity buffer contains 1mM UTP and 1 mM ATP. We have clarified the text in this section to refer to the fact that CtpS is polymerized in complete activity and then additional UTP and ATP are added later to stimulate the depolymerization/repolymerization cycle.

*“CtpS's product, CTP, is both necessary and sufficient to induce CtpS polymerization.” There is no CTP in activity buffer, but polymerization occurs,*
Figure 1*. Ah, I see explained in the response to the original review that the assembly induced in*
Figure 1
*is now thought to be produced entirely by the CTP generated by the CtpS when all substrates are present. Wow. The authors should be aware that I completely missed this in the text. I would suggest warning the reader after*
Figure 1
*that, although it looks like the substrates are inducing assembly, it is actually the product CTP, which is produced only when all substrates are present, that is causing assembly. And then when discussing*
Figure 2
*(and*
Figure 2—figure supplement 1*) the interpretation should be emphasized again and all evidence summarized*.

We thank the reviewer for pointing out the lack of clarity in our writing. We have clarified the Results section to highlight the fact that, though substrates are present when polymerization occurs, we think it is the CTP accumulation that is most relevant to the polymerization.

*This revelation now presents additional questions for*
Figure 1*. Methods state that LS was monitored for 5 minutes, but later “for 3 minutes unless otherwise noted.” But what was the level of CTP generated in this time?*
Figure 2
*shows that 1 mM CTP was needed for maximum assembly (of the unknown concentration of CtpS). Since the substrates were present at 1 mM, this would be the maximum amount possible CTP that could be generated. The question is, how fast does it approach the 0.5-1.0 mM level? This would seem to introduce an additional level of complexity to the kinetics, and it would seem important to see a couple of curves showing the kinetics of CTP production, say at high and low CtpS concentrations. Most important would be to know what level of CTP was produced at the 3 minute time point? Again we would need this for a high and low CtpS. Best to show some kinetic curves in SI*.

We agree that it would be useful to consider the amount of CTP produced during individual experiments. We would like to note that we observe the most polymerization and the majority of the enzymatic activity in the first minute of observation, so this earlier activity seems to have the largest effect. We have included a supplemental figure (Figure 1—figure supplement 2) to show some kinetic curves from individual trials at three one low, medium, and high concentration of CtpS. At 100 nM CtpS (Figure 1—figure supplement 2) the change in transmittance (showing CTP production) is essentially linear, allowing us to calculate the total CTP produced based on the *k*cat of 5.5 s-1, producing approximately 100 uM CTP over the course of the entire assay (3 minutes). Above the threshold for polymerization, most of the overall change in light scattering occurs within the first minute of the reaction, corresponding to the highest enzymatic velocities. The CTP production curves at higher protein concentrations are more complex but we can approximate that 2 uM CtpS (Figure 1—figure supplement 2) produces ∼460 uM CTP over the course of the reaction. In the same time frame, 5 uM CtpS (Figure 1—figure supplement 2) produces approximately 380 uM CTP.

*This also suggests that the attempt to identify a polymerization threshold in 1B may be complicated by the kinetics of CTP formation. It would seem better to shift the search for a polymerization threshold to the conditions in 2A, a clean system with constant CTP. Unfortunately, the LS system has already shown itself inadequate. But how about the centrifugation assay, used only once in 2B. Might this be a better assay to indicate a polymerization threshold*.

We agree with the reviewer that the strongest evidence for a clear polymerization threshold comes from the experiments in 2A with saturating CTP. However, the ultracentrifugation assay is not suitable for this because of the limited linear range of immunoblotting. While the light scattering may not be ideal for determining a precise, defined critical concentration, the point we wish to highlight is that the experiments in 1B and 2A return values that are consistent with one another, providing support that CTP alone is the main factor stimulating polymerization. We have added more explanation for this in the Results section.

*“GTPase assembly is repressed by its substrates” This is also not clear to me. Maybe it is from*
Figure 2*, but that is complicated by the DON inhibitor*.

Our conclusion that CtpS assembly is repressed by substrates is based on Figure 1, where the addition of substrates causes depolymerization and Figure 2 where addition of substrates in the presence of a glutamine analog causes depolymerization without repolymerization.

*The model presented in SI proposes a dimer nucleation step, which is difficult to imagine for a linear polymer where all subunit interfaces are the same. This enigma has been discussed for FtsZ, which is also one subunit thick, and for which cooperative assembly is convincingly established. A possible resolution has been addressed by Miraldi and Romberg Biophys J 95:2470, which the authors should consider*.

In the Results section we suggest that the slight difference in conformation between polymeric CtpS and the crystal structure of CtpS (“filament” and “free” forms describe in the main text) may be responsible for the cooperative assembly, with the filament form favoring polymerization. This could involve a dimerization-induced conformational change as described by Miraldi et al or the free energy change involved in the restriction of lability that we propose. Because both of these are possible, we have included a reference to Miraldi et al in the Results section and expanded our Discussion.

[Editors’ note: the author responses to the first round of peer review follow.]

Reviewer 1

*1) The “activity buffer” apparently came from an earlier paper measuring the enzyme activity. Curiously it contains no monovalent cations. I think it would be important to provide at least a quick test of how assembly and enzyme activity are affected by 300 mM potassium, i.e., in a physiological buffer*.

To address the reviewer’s concern about the physiological relevance of polymerization mediated enzymatic regulation, we have repeated the CtpS titration experiment shown in Figure 1 with activity buffer supplemented with 150 mM KOAc (within the physiological [K+] range measured in Shabala *et al* 2009). We find that the overall trends observed in Figure 1 are maintained in the presence of K+; polymerization and a corresponding decrease in enzymatic activity both occur above the threshold value of 1 uM determined for CtpS polymerization (see Figure 10). Previous CtpS enzymatic activity assays used in the literature typically omit monovalent cations or keep them at low levels, so for consistency with past experiments and internal consistency with other experiments performed in the manuscript, we have not included this data in the main text.Author response image 1.

*2) The existence of a critical concentration would imply a cooperative assembly, and this is an important issue. But I don't think the curve in*
Figure 1
*is convincing. A Cc should be seen as a flat line of zero assembly below Cc, and a straight line of constant slope above. The curve in 1a might be described as a small slope from 0-7 µM, a steeper rise to 10, and approaching a plateau by 20. However the points are erratic and might also be described as a single line from 0-20. I see error bars on only two points. I think it is impossible to judge a mechanism from this erratic LS data, and certainly a Cc is not convincing. The logarithmic plots in S1 don't help, since a Cc is best seen in a linear plot. The Figure caption is sloppy. It indicates a log plot in an inset (but it is in SI), and 1c is confusing. It is concluded that activity increases as the polymers in the pellet disassemble, but the disassembly is not confirmed by any measurement*.

The reviewer raises several important issues in this point and we have addressed all of them.

A) First, it is important to note that the conclusion that CtpS assembles through a cooperative mechanism with a threshold concentration for polymerization is not just based on curve fitting of Figure 1 (now Figure 1 in the revised manuscript). If there were no threshold, then one would always observe more tetramers than dimers of tetramers, more dimers of tetramers than trimers of tetramers, etc. and thus the bulk of protein would be in tetramers and small oligomers of tetramers. The fact that we see such long polymers by EM and such a large fraction of protein in polymers by ultracentrifugation indicate that CtpS assembly is nucleation dependent and that there exists a critical concentration. Figure 1 attempts to measure that threshold concentration for polymerization, but due to the nonlinearity of assembly we are intentionally cautious in interpreting this measurement. Since this caveat was not clear, we have altered the text to refer to an assembly threshold (a qualitative description of the assembly behavior) rather than a critical concentration, which, as the reviewer notes, is associated with a specific quantitative definition. That said, we have made additional activity and assembly measurements at additional CtpS concentrations to improve the continuity of Figure 1 and the new data are consistent with the conclusion that CtpS assembles cooperatively above a threshold of roughly 1 μM.

B) There were error bars on the original graph but they were so small that they were obscured by the large data points. We have altered the formatting of the plot to make them clearer.

C) We have fixed the figure caption.

D) We have demonstrated CtpS disassembly in the experiment of Figure 1 by including a supplemental figure showing the corresponding light scattering change over time (Figure 1—figure supplement 4).

*3) “polymers are inactive or significantly less active.” If there were a Cc, and polymers were inactive, one would expect to see activity increase in direct proportion to CtpS concentration below Cc, and flat-line above Cc. The mechanism is clearly more complex. “CtpS was incubated in activity or CTP buffer [this is not defined] for one hr...centrifuged for 15-30 min.” A science paper should report the exact conditions used for the data shown*.

We thank the reviewer for pointing out this ambiguity and have clarified the conditions used in the Materials and methods section. We agree with the reviewer that the mechanism is clearly more complex than the simplest polymerization model presented, since total activity decreases nonhyperbolically. However, the simple model does capture the ultrasensitive nature of the regulation and is thus a useful framework for the simple analysis in the paper. We have noted the possibility of more complex models in the Discussion.

*4) Polymerization was apparently initiated by adding glutamine, and the monitored for 3 minutes. The initiation by glutamine was referenced to their 2010 paper, but that paper showed no kinetic analysis. We really need a detailed analysis how glutamine induces polymerization – a time course of LS and enzyme activity, as a function of CtpS concentration. This is likely to invite a more complex analysis, including the question of nucleation, but that should be a part of any conclusion about a Cc. At the very least we need to know that the LS and enzyme activity plotted are at steady state.*
Figure 2
*suggest that LS is still increasing at 3-5 minutes*.

We have taken the reviewer’s suggestion and performed experiments that demonstrate that neither glutamine itself nor the glutamine analog, DON, directly impacts CtpS assembly (Figure 2—figure supplement 1).

*5) The negative stain is poorly described: ” in 50% glycerol [no buffer?]...with or without CTP and MgCl2 [don't you need to state which were used for which figures?]...diluted 1/10 in storage buffer [not defined, and why was everything not in the same buffer as the assays?]*.

We have expanded our description of the negative staining procedure in the text.

*6)*
Figure 1
*“sharp decrease in LS”: This may have been caused by agitation of the polymers as the substrate [what was the substrate, there are 3??] was added and mixed. This needs controls of buffer only*.

We have performed the requested control and reach the same conclusions. The new data and text (clarifying that the “substrates” added were UTP and ATP) are presented in Figure 1—figure supplement 5 and discussed in the Results section.

*7)*
Figure 2
*has many of the same serious problems as*
Figure 1*. It raises the observation that polymers can be initiated by glutamine, a substrate, or by CTP, the product, which I thought inhibits in presence of glutamine? This (and the inhibition by ATP and UTP) suggests that we really need a comprehensive study of how the substrates and CTP stimulate and inhibit assemble*.

As discussed in the response to point #4, we have performed a more comprehensive study of how the substrates and CTP stimulate and inhibit assembly (Figure 2—figure supplement 1). We conclude that only CTP stimulates assembly and that nucleotide substrates, UTP and ATP, stimulate disassembly. The observation that assembly is only seen when all of the substrates are present is consistent with the conclusion that the assembly in this case is stimulated by the CTP produced by the enzyme. Consistently, the substrates fail to stimulate assembly of the E155K mutant which is active but deficient in CTP binding (Figure 2). Furthermore, the wild type enzyme fails to assemble when CTP production is inhibited by DON, but DON-inhibited CtpS still polymerizes in the presence of added CTP (Figure 2; Figure 2—figure supplement 1 and Figure 2—figure supplement 5).

We also provide new experimental support for the idea that CTP stabilizes the filament by binding more tightly to it than to the tetramer; since CTP has higher apparent affinity to CtpS at protein concentrations that promote polymerization, and lower affinity to a mutant CtpS with a polymer-disrupting mutation (Figure 7—figure supplement 2). In turn, we demonstrate that the presence of high concentrations of CTP promote polymerization (Figure 8—figure supplement 3). Both of these results agree with predictions from thermodynamic linkage between CTP binding and polymerization in which CTP preferentially binds the polymer.

*8) “…indicating that tetramer formation precedes filament formation.” This throws another major variable into the story. When one starts with CtpS in whatever buffer, is it a monomer or tetramer? Do substrates cause tetramer formation? CTP? If tetramers need to form before filaments, this will greatly complicate the assembly mechanism*.

Previous studies have dissected CtpS tetramerization in great detail. All of our results are consistent with the conclusion that CtpS polymerization reflects tetramer-tetramer assembly. To enable us to focus specifically on the tetramer-tetramer interactions, all of the assembly experiments were performed by pre-assembling tetramers with ATP and UTP (glutamine is not necessary for tetramerization) and then initiating enzymatic activity with glutamine addition, which we independently demonstrated does not stimulate assembly independently of CTP production (see response to point #7 above).

To help the reader think about dimer to tetramer to polymer transitions, we have included model figures at the beginning of the paper and at the end, each of which depict the nucleotide dependences for the different oligomeric forms (Figure 1, Figure 9).

*9) The cryoEM can't really be judged by non-experts, but a couple of things caught my attention. It is first stated that the filament is fit by x-shaped CtpS tetramers. Then in 4b the structure is fit by CtpS monomers, with a (apparently flexible) linker between the two domains. What happened to the tetramers, and how do the monomer domains in the final fitting compare to their position in the tetramers? OK, I see in 6A that the fitted (flexible) monomers seem to superimpose almost exactly on the tetramer. I don't understand the conclusion ”a compression of the tetramer along the filament axis [not defined, but maybe vertical on the figure?]. The superposition seems to me extremely precise*.

The shift at the tetramerization interface is indeed quite small, and we have edited the text to indicate that it results in a compression of about 3 Å along the helical axis.

*10) “There is strong density for CTP at the inhibitory site.” Is it really possible for a single CTP to produce strong density in a 10 Å cryoEM reconstruction? Even more so for the phosphates of ADP, and the missing density of the base - how could the base be partially disordered? The conditions for making the polymers for cryoEM are not given, but it seems CTP was used*.

We have clarified in the text how the filaments were generated for cryo-EM imaging. It is, in fact, possible to see clear density for CTP and ADP in the reconstruction; we have clarified this by including in Figure 3—figure supplement 2 a difference map between the EM structure and a calculated model that lacks CTP and ADP. This map shows strong peaks (>8 sigma) at the positions predicted for CTP and ADP, but none in the predicted Gln and GTP binding sites. This represents even stronger evidence than our original analysis that the CTP and ADP sites are occupied. We have edited the text to reflect this.

*11)*
Figure 8
*shows the speculative identification of 4 aa's that are on the linker and that might be part of a subunit interface. These were mutated, and all of the destroyed the* in vivo *localization – the mCherry is diffuse in the cytoplasm. However, the mCherry seems to be at the N terminus, and it could fold even if the fused CtpS were defective.*
Figure 9
*provides some support for the suggestion that E277R does block polymerization and is accompanied by increased enzyme activity at the higher CtpS concentrations, where wt protein apparently loses activity upon polymer formation. If this were coupled with a more comprehensive study of kinetics of assembly, it would be an interesting if not definitive finding*.

The in vitro enzymatic activity data show that E277R can make CTP and in vivo*,* mutants with E277R as the sole copy of CtpS can grow in minimal media while *pyrG* (*E. coli* CtpS) null mutants cannot. Both of these results indicate that E277R’s overall structure is properly folded and the enzyme can produce CTP in the cell.

Reviewer 2

*[…] This is an elegant study. The manuscript is one of the best manuscripts that I have reviewed in many months. The data are convincing and the authors have not made any major assertions, which their results do not seem to back up*.

We thank the reviewer for these very kind words.

Reviewer 3

1) Are there error bars in all plots that are in some cases invisible? If not, would be good to add, if so, would be good to change from the squares and diamonds to small points so that error bars are more visible.

All plots have error bars as indicated and we have reformatted them to make them more visible.

2) Is it the case that the CTPs mutant cells in Figure 8 are smaller? They appear so to me; it would be useful to quantify this, and if this holds up, this is further evidence that the mutants are impacted in growth.

We have measured the length and width of these cells and do not see a statistically significant difference from WT. The data are included below (Figure 11) but we did not feel that they were necessary for the paper.Author response image 2.

*3) I was confused as to why they predicted that a multiple site mutant would have exacerbated effects*; *might this not be true if polymerization is already abolished in E277R?*

The reviewer raises a good point and we have removed this prediction.